# CTBench: A Library and Benchmark for Certified Training

## Abstract

Training certifiably robust neural networks is an important but challenging task. While many algorithms for (deterministic) certified training have been proposed, they are often evaluated on different training schedules, certification methods, and systematically under-tuned hyperparameters, making it difficult to compare their performance. To address this challenge, we introduce CTBench, a unified library and a high-quality benchmark for certified training that evaluates all algorithms under fair settings and systematically tuned hyperparameters. We show that (1) almost all algorithms in CTBench surpass the corresponding reported performance in literature in the magnitude of algorithmic improvements, thus establishing new state-of-the-art, and (2) the claimed advantage of recent algorithms drops significantly when we enhance the outdated baselines with a fair training schedule, a fair certification method and well-tuned hyperparameters. Based on CTBench, we provide new insights into the current state of certified training and suggest future research directions. We are confident that CTBench will serve as a benchmark and testbed for future research in certified training.

## 1 Introduction

As a crucial component of trustworthy artificial intelligence, adversarial robustness (Szegedy et al., 2014; Goodfellow et al., 2015), *i.e.*, resilience to small input perturbations, has established itself as an important research area. While initially the community focused on heuristic methods to craft adversarial examples and defenses against these, it turned out that such defenses are often brittle and can be evaded by adaptive adversaries (Athalye et al., 2018; Tramèr et al., 2020). Thus, neural network certification has emerged as a method for providing provable guarantees on the robustness of a given network (Gehr et al., 2018; Wong & Kolter, 2018; Zhang et al., 2018; Singh et al., 2019).

Two families of neural network certification methods have been proposed: complete methods (Katz et al., 2017; Tjeng et al., 2019) which compute the exact bounds but are extremely computationally expensive, and convex-relaxation based methods (Zhang et al., 2018; Singh et al., 2019) which provide approximate bounds but are more scalable. State-of-the-art (SOTA) verifiers (Xu et al., 2021; Ferrari et al., 2022; Zhang et al., 2022) combine both approaches, by using convex relaxations to speed up the solving of complete methods via Branch-and-Bound (Bunel et al., 2020).

However, the scalability of neural network certification is still a major challenge since the computational complexity of SOTA verifiers grows exponentially with network size. To tackle this issue, certified training (Mirman et al., 2018; Gowal et al., 2018) was proposed in order to train neural networks that are amenable to certification. Such methods are typically categorized into two groups: (1) training with a sound upper bound of the robust loss (Zhang et al., 2020; Shi et al., 2021), and (2) training with an unsound surrogate loss that aims to approximate the exact robust loss (Müller et al., 2023; Mao et al., 2023; De Palma et al., 2024). The latter group has been shown to be more effective.

While certified training has made significant advances, there is currently no benchmark that can be used to fairly evaluate the effectiveness of the different certified training methods. Specifically, the literature often compares against previous methods using quoted numbers due to high computational costs, although the verifier and certification budget differ. These unfair comparisons ultimately hinder the community from drawing reasonable conclusions on the effectiveness of certified training methods. In addition, existing works systematically under-tune hyperparameters, in order to show

effectiveness against baselines, thus establishing a weaker SOTA. Further, there is no unified codebase for these methods, making future development and comparison difficult.

**This work: a unified library and high-quality benchmark for certified training** We address these challenges, for the first time unifying SOTA certified training methods into a single codebase called CTBENCH. This enables a fair comparison between certified training methods and re-establishes a much stronger SOTA by fixing problematic implementations and systematically tuning hyperparameters. As shown in Figure 1, these steps lead to significant improvements uniformly. In addition, we show that the claimed advantage of recent SOTA reduces significantly when we apply the same budget and hyperparameter tuning to all methods. Based on

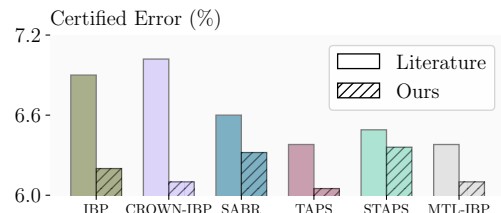

Figure 1: Reduction in certified error on MNIST $\epsilon = 0.3$ (lower is better).

our released model checkpoints, we provide an extensive analysis of the model properties, highlighting many new insights on its loss landscape, mistake patterns, regularization strength, model utilization, and out-of-distribution generalization. We are confident that CTBENCH will serve as a benchmark and testbed for future work in certified training.

## 2  RELATED WORK

We now briefly review key developments most related to our work.

**Benchmarking Certified Robustness** Li et al. (2023) provides the first benchmark for certified robustness, covering not only deterministic certified training but also randomized certified training and certification methods. However, it is outdated and thus provides little insight into the current SOTA methods. For example, it reports 89% and 51% best certified accuracy for MNIST $\epsilon = 0.3$ and CIFAR-10 $\epsilon = \frac{2}{255}$ in its evaluation, respectively, while recent methods have achieved more than 93% and 62% (Müller et al., 2023; Mao et al., 2023; De Palma et al., 2024).

**Certified Training** DIFFAI (Mirman et al., 2018) and IBP (Gowal et al., 2018) apply box relaxation to upper bound the worst-case loss for training. Efforts have been made towards applying more precise approximations: Wong et al. (2018) and Balunovic & Vechev (2020) apply DEEPZ (Singh et al., 2018), while Zhang et al. (2020) incorporate linear relaxations (Zhang et al., 2018; Singh et al., 2019). While these approximations are more precise (Baader et al., 2024), they often lead to worse training results, attributed to non-smoothness (Lee et al., 2021), discontinuity and sensitivity (Jovanović et al., 2022) of the loss surface. Some recent work (Balauca et al., 2024) aim to mitigate these problems, however, the most effective training approximation is still the least precise box relaxation. In this regard, the focus of the community has shifted towards improving IBP: Shi et al. (2021) propose a new regularization and initialization paradigm to speed up IBP training; De Palma et al. (2022) apply IBP regularization to make adversarial training certifiable; Müller et al. (2023), Mao et al. (2023) and De Palma et al. (2024) propose unsound but more effective IBP-based surrogate losses for training; Mao et al. (2024) propose to use wider models instead of deeper models for IBP-based methods. These methods achieve universal advantages and are thus the focus of our work.

## 3  BACKGROUND

We now introduce the necessary background for our work, both training concepts and algorithms.

### 3.1  TRAINING FOR ROBUSTNESS

We present the mathematical notations on adversarial and certified training here. We consider a neural network classifier $f_\theta(x)$ that estimates the log-probability of each class and predicts the class with the highest estimated log-probability.

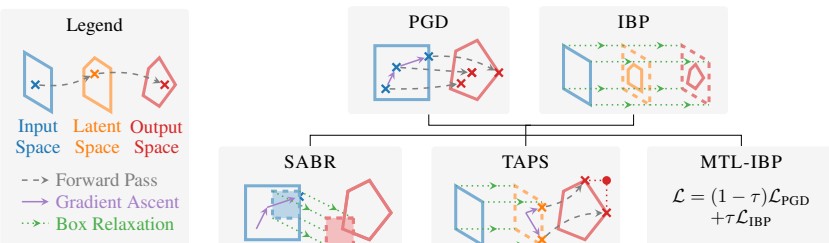

Figure 2: Conceptual overview of core algorithms built into CTBENCH.

**Adversarial Training** A classifier $f_\theta(x)$ is said to be *adversarially robust* with radius $\epsilon$ w.r.t. $L_\infty$ perturbation if $f_\theta(x + \delta) = y$ for all $\|\delta\|_\infty \leq \epsilon$, where $y$ is the ground truth label of $x$. For brevity, we will omit the perturbation type in the rest of the paper. Finding an adversarially robust classifier is formally defined to solve a min-max problem $\theta = \arg\min_\theta \mathbb{E}_{x,y} \max_{\|\delta\| \leq \epsilon} L(x + \delta)$. In this regard, adversarial training solves the inner maximization problem by generating adversarial examples during training, and the outer minimization problem by optimizing the empirical loss of adversarial examples.

**Certified Training** A classifier $f_\theta(x)$ is said to be *certifiably robust* if it is adversarially robust and there exists a sound verifier that certifies the robustness. A verifier typically computes an upper bound on the margin $f_i(x + \delta) - f_y(x + \delta)$ and certifies its robustness if the upper bound is negative for all $i \neq y$. Certified training thus replaces the inner maximization problem with an upper bound and minimizes the upper bound during training instead.

**Metrics** The main metric for certified training is *certified accuracy*, defined to be the ratio of certifiably robust samples in the dataset. The ratio of correctly classified samples in the dataset is thus called *natural accuracy*. For reference, we include *adversarial accuracy* as well, defined to be the ratio of adversarially robust samples in the dataset. We apply one of the most widely used SOTA certification methods, MN-BAB (Ferrari et al., 2022), as the verifier. To compute adversarial accuracy, we apply the strong AUTOATTACK (Croce & Hein, 2020) for adversarial training, and a combination of PGD attack and branch-and-bound attack from MN-BAB for certified training.

### 3.2 ALGORITHMS IN CTBENCH

Here, we briefly introduce the core algorithms built into CTBENCH. Concepts behind these algorithms are visualized in Figure 2.

**PGD and EDAC** Projected Gradient Descent (PGD) (Madry et al., 2018) is the most widely recognized adversarial training method. Starting from a random initialization, PGD solves the inner maximization problem by iteratively taking a step towards the gradient direction and clipping the result into the valid perturbation set. Then, it uses the generated adversarial input $x'$ to compute the worst case loss as $L(x')$. Croce & Hein (2020) find that PGD remains effective against strong attacks, thus is popular as an integrated part of many certified training methods (Müller et al., 2023; Mao et al., 2023; De Palma et al., 2024). To further improve adversarial robustness, Zhang et al. (2023) improves adversarial generalization via an extra-gradient method called EDAC, which remains one of the SOTA methods in adversarial training. These methods achieve good but uncertifiable adversarial robustness, hence we use them as adversarial robustness baselines in CTBENCH.

**IBP** Interval Bound Propagation (IBP) (Gowal et al., 2018) uses interval analysis to approximate the output range of each layer. For example, for the toy network $y = 2 - \text{ReLU}(x_1 + x_2)$ with input bounds $x_1, x_2 \in [-1, 1]$, it first computes the output range of the first layer as $x_1 + x_2 \in [-1, 1] + [-1, 1] \subseteq [-2, 2]$, the second layer as $\text{ReLU}([-2, 2]) \subseteq [0, 2]$ and then final layer as $2 - [0, 2] \subseteq [0, 2]$, thus proving $y \geq 0$ for all possible $x_1, x_2 \in [-1, 1]$. Similarly, IBP computes the layer-wise bounds and then derives the worst-case loss based on the output bounds of the final layer. To stably train models with IBP, Shi et al. (2021) propose to rescale the parameter initialization to ensure constant growth of IBP bounds and a specialized regularization to control the activation

status of neurons. They also show that adding a batch norm (Ioffe & Szegedy, 2015) layer before every ReLU can improve IBP training. These training tricks are adopted by every IBP-based method introduced below. For brevity, we refer to this variant as IBP in the rest of the paper unless otherwise stated, since it improves the original IBP universally with tricks that facilitate training.

**CROWN-IBP**  CROWN-IBP (Zhang et al., 2020) tightens the imprecise interval analysis with linear relaxations of ReLU layers based on IBP bounds and only solves the linear constraints for the final layer output based on CROWN (Zhang et al., 2018), avoiding prohibitive costs during training. To further reduce the cost of solving the bounds for each class, Xu et al. (2020) propose a loss fusion trick to only solve for the final loss, thus reducing the asymptotic complexity by a factor equal to the number of classes. For brevity, we refer to this variant as CROWN-IBP in the rest of the paper unless otherwise stated, since the original CROWN-IBP cannot scale to datasets with many classes.

**SABR**  Since IBP is often criticized for the increasingly strong regularization w.r.t. input radius imposed on the neural network, SABR (Müller et al., 2023) proposes to use IBP only for a carefully chosen small box inside the original input box for IBP training. More specifically, it first conducts a PGD attack in the full input box to find an adversarial input, and then takes the surrounding small box with radius $\lambda\epsilon$ around the adversarial input as the input box for IBP training, where $\lambda$ is a pre-defined ratio. For exceptional cases (specifically CIFAR-10 $\epsilon = \frac{2}{255}$), SABR further shrinks the output box of every ReLU towards zero by a pre-defined constant to further reduce the regularization.

**TAPS and STAPS**  Observing that IBP relaxation error grows exponentially w.r.t. model depth (Müller et al., 2023; Mao et al., 2024), TAPS (Mao et al., 2023) proposes to split the network into two subparts, using IBP for the first subpart and PGD for the other. This way, the over-approximation from IBP and the under-approximation from PGD partially cancel out, yielding a more precise approximation of the worst-case loss. Further, TAPS uses a separate PGD attack to estimate the bounds for every class to align better with the certification objective. STAPS (Mao et al., 2023) combines TAPS with SABR by using the adversarial small box for TAPS training, thus further reducing regularization.

**MTL-IBP**  De Palma et al. (2024) formalizes a family of surrogate loss functions that interpolate between PGD and IBP training. We study MTL-IBP, one of the most effective algorithms in this family. MTL-IBP linearly interpolates between PGD loss and IBP loss, *i.e.*, $\mathcal{L} = (1 - \tau)\mathcal{L}_{\text{PGD}} + \tau\mathcal{L}_{\text{IBP}}$, where $\tau$ is the pre-defined IBP coefficient. To recover the re-weighing between PGD and IBP as SABR does with box shrinking, MTL-IBP uses a larger input radius for a PGD attack in the same setting (specifically, CIFAR-10 $\epsilon = \frac{2}{255}$).

## 4  A UNIFIED LIBRARY AND HIGH-QUALITY BENCHMARK FOR CERTIFIED TRAINING

We now discuss CTBENCH, both the unified library and the corresponding benchmark.

### 4.1  THE CTBENCH LIBRARY

We implement every algorithm described in Section 3.2 in a unified framework. The training loss is composed of three components: the natural loss which measures performance on clean inputs, the robust loss which measures robust performance depending on the concrete algorithms and regularization losses which are used to stabilize training and improve generalization. Formally, the training loss is defined as $\mathcal{L} = (1 - w_{\text{rob}})\mathcal{L}_{\text{nat}} + w_{\text{rob}}\mathcal{L}_{\text{rob}} + \mathcal{L}_{\text{reg}}$. We mainly use $L_1$ regularization to reduce overfitting and the warmup regularization proposed by Shi et al. (2021) to improve certified training methods. The IBP initialization (Shi et al., 2021) is applied for every certified training method, while adversarial training is initialized with Kaiming uniform (He et al., 2015). Every method has a warmup phase where $\epsilon$ is increased from 0 to the target value and a fine-tuning phase where the model continues to train at the targeted $\epsilon$ to converge. The learning rate is held constant during the warmup phase and decayed in the fine-tuning phase with a constant multiplier. We use CNN7 as the model architecture, in agreement with recent literature (Shi et al., 2021; Müller et al., 2023; Mao et al., 2023; De Palma et al., 2024).

Table 1: CTBENCH results with comparison to the literature. We include the natural accuracy of standard training and adversarial training, and the adversarial accuracy of adversarial training for reference. The best numbers are in bold and those exceeding the literature results are underlined.

| Dataset Std. Nat. Accu. | $\epsilon_\infty$ | Training Method | Source | Nat. [%] Literature | Nat. [%] CTBench | Cert. [%] Literature | Cert. [%] CTBench | Adv. [%] CTBench |
|---|---|---|---|---|---|---|---|---|
| | 0.1 | PGD | / | / | 99.47 | / | $\approx 0^\dagger$ | 98.97 |
| | | EDAC | / | / | 99.58 | / | $\approx 0^\dagger$ | 98.95 |
| | | IBP | Shi et al. (2021) | 98.84 | 98.87 | 97.95 | 98.26 | 98.27 |
| | | CROWN-IBP | Xu et al. (2020) | 98.83 | 98.94 | 97.76 | 98.21 | 98.23 |
| | | SABR | Müller et al. (2023) | 99.23 | 99.08 | 98.22 | 98.40 | 98.47 |
| | | TAPS | Mao et al. (2023) | 99.19 | 99.16 | **98.39** | **98.52** | 98.58 |
| MNIST | | STAPS | Mao et al. (2023) | 99.15 | 99.11 | 98.37 | 98.47 | 98.50 |
| | | MTL-IBP | De Palma et al. (2024) | **99.25** | 99.18 | 98.38 | 98.37 | 98.44 |
| 99.50 | 0.3 | PGD | / | / | 99.43 | / | $\approx 0^\dagger$ | 93.83 |
| | | EDAC | / | / | 99.51 | / | $\approx 0^\dagger$ | 95.02 |
| | | IBP | Shi et al. (2021) | 97.67 | 98.54 | 93.10 | 93.80 | 94.30 |
| | | CROWN-IBP | Xu et al. (2020) | 98.18 | 98.48 | 92.98 | 93.90 | 94.29 |
| | | SABR | Müller et al. (2023) | 98.75 | 98.66 | 93.40 | 93.68 | 94.46 |
| | | TAPS | Mao et al. (2023) | 97.94 | 98.56 | **93.62** | **93.95** | 94.66 |
| | | STAPS | Mao et al. (2023) | 98.53 | 98.74 | 93.51 | 93.64 | 94.36 |
| | | MTL-IBP | De Palma et al. (2024) | **98.80** | 98.74 | **93.62** | 93.90 | 94.55 |
| | $\frac{2}{255}$ | PGD | / | / | 88.67 | / | $\approx 0^\dagger$ | 72.41 |
| | | EDAC | / | / | 89.18 | / | $\approx 0^\dagger$ | 72.42 |
| | | IBP | Shi et al. (2021) | 66.84 | 67.49 | 52.85 | 55.99 | 56.10 |
| | | CROWN-IBP | Xu et al. (2020) | 71.52 | 67.60 | 53.97 | 57.11 | 57.28 |
| | | SABR | Müller et al. (2023) | 79.24 | 77.86 | 62.84 | 63.61 | 65.56 |
| | | TAPS | Mao et al. (2023) | 75.09 | 74.44 | 61.56 | 61.27 | 62.62 |
| CIFAR-10 | | STAPS | Mao et al. (2023) | 79.76 | 77.05 | 62.98 | 64.21 | 66.09 |
| | | MTL-IBP | De Palma et al. (2024) | **80.11** | **78.82** | 63.24 | **64.41** | 67.69 |
| 91.27 | $\frac{8}{255}$ | PGD | / | / | 78.71 | / | $\approx 0^\dagger$ | 35.93 |
| | | EDAC | / | / | 78.95 | / | $\approx 0^\dagger$ | 42.48 |
| | | IBP | Shi et al. (2021) | 48.94 | 48.51 | 34.97 | 35.28 | 35.48 |
| | | CROWN-IBP | Xu et al. (2020) | 46.29 | 48.25 | 33.38 | 32.59 | 32.77 |
| | | SABR | Müller et al. (2023) | 52.38 | 52.71 | 35.13 | 35.34 | 36.11 |
| | | TAPS | Mao et al. (2023) | 49.76 | 49.96 | 35.10 | 35.25 | 35.69 |
| | | STAPS | Mao et al. (2023) | 52.82 | 51.49 | 34.65 | 35.11 | 35.54 |
| | | MTL-IBP | De Palma et al. (2024) | **53.35** | **54.28** | 35.44 | 35.41 | 36.02 |
| | $\frac{1}{255}$ | PGD | / | / | 46.78 | / | $\approx 0^\dagger$ | 33.16 |
| | | EDAC | / | / | 46.79 | / | $\approx 0^\dagger$ | 33.16 |
| | | IBP | Shi et al. (2021) | 25.92 | 26.77 | 17.87 | 19.82 | 19.84 |
| TINYIMAGENET | | CROWN-IBP | Xu et al. (2020) | 25.62 | 28.44 | 17.93 | 22.14 | 22.31 |
| | | SABR | Müller et al. (2023) | 28.85 | 30.58 | 20.46 | 20.96 | 21.16 |
| 47.96 | | TAPS | Mao et al. (2023) | 28.34 | 28.64 | 20.82 | 21.58 | 21.71 |
| | | STAPS | Mao et al. (2023) | 28.98 | 30.63 | 22.16 | 22.31 | 22.57 |
| | | MTL-IBP | De Palma et al. (2024) | **37.56** | **35.97** | **26.09** | **27.73** | 28.49 |

† None of the first 10 samples are certified due to the time limit of 1000 seconds per sample.

Due to the importance of batch norm in certified training, we consider it as a native part of CTBENCH. Specifically, the best practice so far is to set batch norm statistics based on the clean input and use this for computing IBP bounds. However, we find several problematic implementations of batch norm in the literature: (1) when gradient accumulation is involved, the batch norm statistics are not updated correctly, as sub-batch statistics are applied for training; (2) batch norm statistics change more than once before taking a gradient step, as typically running statistics is used for conducting a PGD attack and thus evaluating $\mathcal{L}_{\text{rob}}$, while $\mathcal{L}_{\text{nat}}$ is evaluated with batch statistics. The first problem makes gradient accumulation ineffective since the quality of batch statistics depends highly on the batch size, and the second problem prevents training with $w_{\text{rob}} \in (0, 1)$. To address the first problem, we propose to use full batch statistics during gradient accumulation, which leads to slim overheads but allows arbitrary gradient accumulation. To address the second problem, we conduct a PGD attack with the batch statistics as well and evaluate everything with the current batch statistics. This way, the batch norm statistics are set once per batch just like standard training, allowing training with the combination of $\mathcal{L}_{\text{nat}}$ and $\mathcal{L}_{\text{rob}}$. Further, Wu & Johnson (2021) find that running statistics lag behind the population statistics and propose to use the population statistics for testing. We adopt this strategy in CTBENCH, since it only needs to compute $\mathcal{L}_{\text{nat}}$ and is much cheaper than the computation of $\mathcal{L}_{\text{rob}}$.

We find that models trained with the hyperparameters reported in the literature frequently show strong overfitting patterns. To remediate this, we conduct a magnitude search for $L_1$ regularization until the train and validation performance roughly match. To further aid generalization, we apply Stochastic Weight Averaging (Izmailov et al., 2018) for methods that cannot provide metrics for model selection, e.g., MTL-IBP. A more detailed description of the implementation can be found in App. B.

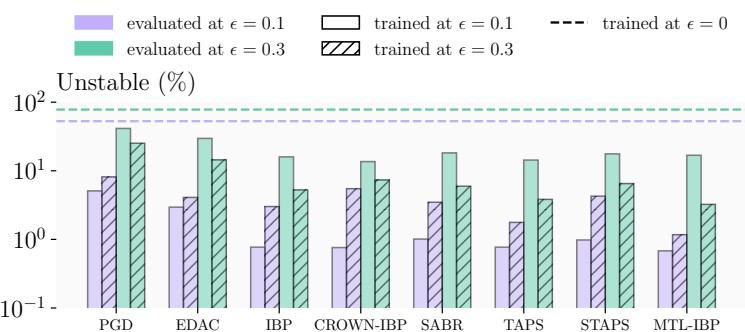

Figure 3: Ratio of unstable neurons for models trained on MNIST with different methods and $\epsilon$.

## 4.2 THE CTBENCH BENCHMARK

Table 1 shows the result of CTBENCH using the methodology described in Section 4.1. We find that CTBENCH achieves consistent improvements in both certified and natural accuracies. In particular, it establishes new SOTA by a margin matching algorithmic advances everywhere except CIFAR-10 $\epsilon = \frac{8}{255}$, where we have $0.03\%$ lower certified accuracy compared to De Palma et al. (2024) but $0.93\%$ higher natural accuracy. This proves the effectiveness of our implementation and the importance of setting batch norm statistics properly in certified training. We also observe the following: (1) when $\epsilon$ is large, the claimed advantage of recent SOTA over IBP drops significantly, from $7.54\%$ relative certified error reduction to $2.42\%$ on MNIST $\epsilon = 0.3$ and from $1.34\%$ relative increase in certified accuracy to $0.45\%$ on CIFAR-10 $\epsilon = \frac{8}{255}$; (2) when the model has sufficient capacity, $e.g.$, on MNIST $\epsilon = 0.1$, certified training can get close to the natural accuracy of standard training ($99.18\%$ for MTL-IBP vs $99.50\%$ for standard training), and they also get similar adversarial accuracy to adversarial training ($98.58\%$ for TAPS vs $98.95\%$ for EDAC), with boosted certified accuracy ($98.52\%$ for TAPS vs almost $0\%$ for EDAC); (3) when $\epsilon$ is large, certified training even gets better adversarial accuracy than PGD training ($94.66\%$ for TAPS vs $93.83\%$ for PGD on MNIST $\epsilon = 0.3$ and $36.11\%$ for SABR vs $35.93\%$ for PGD on CIFAR-10 $\epsilon = \frac{8}{255}$), but there is still a gap between the adversarial accuracy of the SOTA adversarial training methods and that of the SOTA certified training methods, as well as natural accuracy.

## 5 EVALUATING AND UNDERSTANDING CERTIFIED MODELS

We now preform an extensive evaluation on models trained with CTBENCH, providing insights into the current state of certified training. Further experimental results are provided in App. C.

## 5.1 LOSS FRAGMENTATION

ReLU networks are known to have a fragmented loss surface, due to the activation switch of neurons. Fragmentation leads to a non-smooth loss surface and increases the difficulty of finding the worst-case loss via gradient-based methods like PGD. Due to its connection to adversarial robustness, in this section, we investigate the fragmentation of loss surfaces in certified models. Specifically, we answer: (1) do certified models have less fragmentation, thus easing adversarial search, and (2) how does the fragmentation change w.r.t. $\epsilon$?

Fragmentation is closely related to the number of unstable neurons, $i.e.$, neurons that switch activation status in the neighborhood, as all fragments are defined by a group of unstable neurons. Vice versa, in most cases, a switching neuron introduces at least one fragmentation since every activation pattern defines a local linear function. Therefore, we can quantify the fragmentation by the ratio of unstable neurons. Since the exact ratio is NP-complete to compute, we use a heuristic but effective method to estimate it: first, a group of inputs is sampled in the input box; second, these inputs are fed into the model to get the corresponding activation pattern; finally, we count the ratio of unstable neurons observed in the sampled activations. This method always establishes a lower bound of the true ratio

Table 2: Observed count of common mistakes of models on MNIST against their expected values assuming independence across model mistakes.

| | | # models succeeded | | | | | |
|---|---|---|---|---|---|---|---|
| | | 0 | 1 | 2 | 3 | 4 | 5 | 6 |
| $\epsilon = 0.1$ | obs. | 93 | 25 | 21 | 30 | 32 | 56 | 9743 |
| | exp. | 0 | 0 | 0 | 1 | 37 | 900 | 9062 |
| $\epsilon = 0.3$ | obs. | 452 | 73 | 53 | 51 | 80 | 111 | 9180 |
| | exp. | 0 | 0 | 2 | 39 | 445 | 2698 | 6816 |

and gets arbitrarily close when sample size is large enough. In our experiments, we sample the noise 50 times from a standard Gaussian clipped to $[-1, 1]$ and rescale it by $\epsilon$. This sampling focuses more on the neighborhood of the clean input and the boundary of the input box, where new fragments appear most likely. We find this sampling process extremely effective, as the ratio of unstable neurons observed is very close to the upper bounds derived by IBP for certified models.

Figure 3 shows the result of certified models trained at $\epsilon = 0.1$ and $\epsilon = 0.3$ on MNIST, respectively. We evaluate the fragmentation of every model at both $\epsilon = 0.1$ and $\epsilon = 0.3$. First, we observe that both adversarial training and certified training greatly reduce loss fragmentation compared to standard training. Second, comparing different training methods within each group of □ and ▨, we observe that certified training consistently has significantly less fragmentation than adversarial training, *e.g.*, IBP reduces fragmentation by 3x compared to EDAC, thus finding the worst-case loss is much easier. This is consistent with the practice where a weak single-step attack is adopted in certified training (De Palma et al., 2024). Third, comparing models trained at different $\epsilon$ (□ vs ▨ and □ vs ▨), we observe that further increasing training $\epsilon$ does not necessarily reduce fragmentation, yet the trend is consistent with adversarial training. These observations prove that certified training can further boost the fragmentation reduction effect of adversarial training, thus introducing more local smoothness into the model. More results on CIFAR-10 are included in App. C as Figure 7.

## 5.2 SHARED MISTAKES

We now study the correlation of certified models, specifically: do certified models make shared mistakes?

We consider models trained by six certified training methods on MNIST at $\epsilon = 0.1$ and $\epsilon = 0.3$ and calculate the distribution of common mistakes they make. Specifically, we count the number of models that fail to achieve certified robustness for each sample in the test set containing 10k samples. The observed value is compared with the expected value, defined as the number of failed models when models with the same certified accuracy make mistakes independently (rounded to integer if necessary). The result is shown in Table 2. Accordingly, certified models make many shared mistakes, as the number of samples that cannot be certified robust by any of these models systematically exceeds the expected value by a large margin. In addition, the number of inputs that are certified robust by all six models is much larger than the corresponding expected value. These facts suggest that there could be an intrinsic difficulty score for each input, thus curriculum learning (Bengio et al., 2009; Ionescu et al., 2016) could be a promising direction to improve certified training. More results on CIFAR-10 are included in App. C as Table 10. We note that common mistakes are also observed across different certification methods, as shown in Table 9 in App. C.

## 5.3 MODEL UTILIZATION

Model utilization represents how much the model capacity is utilized for the task. Since certified training applies IBP bounds, they systematically deactivate neurons (Shi et al., 2021) to gain precision. However, it is not yet clear whether more advanced certified training methods deactivate fewer neurons, thus utilizing the model capacity better.

We define model utilization to be the ratio of neurons activated by the clean input. Figure 4 visualizes the result for models trained on MNIST at $\epsilon = 0.1$ and $\epsilon = 0.3$. Surprisingly, we find that more advanced certified training methods, TAPS and MTL-IBP, deactivate more neurons than IBP on MNIST $\epsilon = 0.1$, while keeping better natural and certified accuracy. More interestingly, these

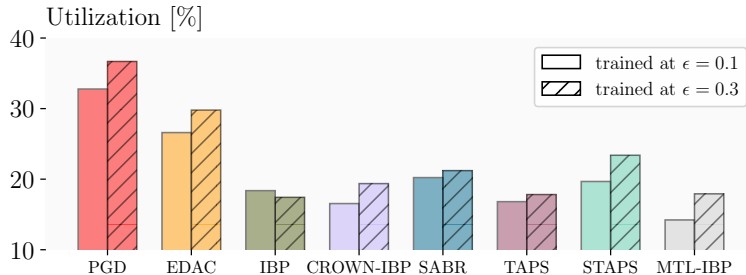

Figure 4: Model utilization for models trained on MNIST with different methods and $\epsilon$. We note that standard training has 42.99% utilization.

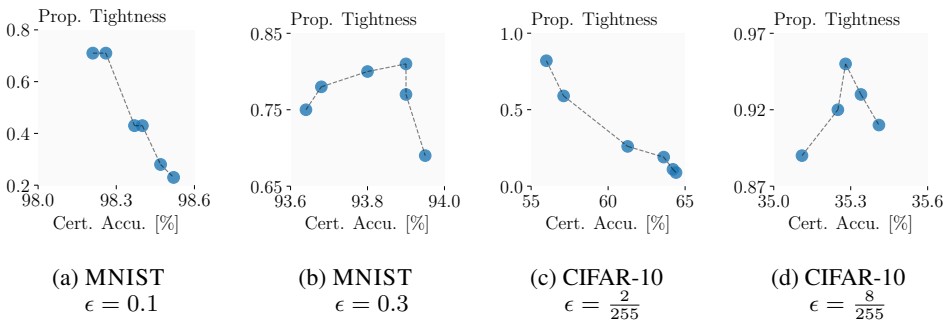

| (a) MNIST | (b) MNIST | (c) CIFAR-10 | (d) CIFAR-10 |
|---|---|---|---|
| $\epsilon = 0.1$ | $\epsilon = 0.3$ | $\epsilon = \frac{2}{255}$ | $\epsilon = \frac{8}{255}$ |

Figure 5: Certified accuracy vs. propagation tightness for models trained on MNIST and CIFAR-10.

methods can retain more utilization on $\epsilon = 0.3$ where the model struggles to keep high natural accuracy for better performance, while IBP has trouble with activating more neurons. Further, we observe that the advanced adversarial training method EDAC shows similar behavior to TAPS and MTL-IBP, and gets higher adversarial accuracy than PGD. This suggests that the ability to adaptively keep necessary utilization could be crucial to both adversarial and certified robustness. Since dying neurons (Lu et al., 2019) are hard to activate again, future work on better warmup (Shi et al., 2021) could be beneficial, as their IBP variant still struggles to keep necessary model utilization. More results on CIFAR-10 are included in App. C as Figure 8.

## 5.4 REGULARIZATION STRENGTH

Previous work (Mao et al., 2024) has shown that IBP bounds are close to optimal bounds for IBP-based certified training, and this condition is established via strong constraints on the model parameters. They quantify this regularization effect by *propagation tightness*, defined to be the ratio between the optimal bound radius and the IBP bound radius, approximating the ReLU network locally with a linear replacement. We now extend the study of propagation tightness to more advanced certified training methods and investigate how it interacts with certified accuracy. Specifically, using propagation tightness as the representative of regularization strength, we answer: (1) do more advanced certified training methods reduce the regularization strength, and (2) how does the input radius $\epsilon$ affect the interaction?

Figure 5 shows the interaction between certified accuracy and propagation tightness for certified models trained on MNIST and CIFAR-10. When $\epsilon$ is small (Figure 5a and Figure 5c), certified accuracy has a negative correlation with propagation tightness, *i.e.*, more advanced certified training methods reduce the regularization strength. However, when $\epsilon$ is large (Figure 5b and Figure 5d), the correlation is not clear, and the best model in certified accuracy does not necessarily have the lowest propagation tightness. Instead, models with similar propagation tightness can have significantly different certified accuracy. Therefore, we conclude that reducing regularization strength cleverly is crucial for certified training, and the effect is more pronounced when $\epsilon$ is small, while improper reduction could hurt certified accuracy, especially when $\epsilon$ is large.

## 5.5 Out-of-Distribution Generalization

Out-of-distribution (OOD) generalization is closely related to adversarial robustness (Gilmer et al., 2019). However, the interaction between certified robustness and OOD generalization is not yet clear. We thus investigate the OOD generalization of certified models and answer: (1) do certified models generalize to OOD data, and (2) how does this compare to adversarial training?

We use MNIST-C (Mu & Gilmer, 2019) to evaluate OOD generalization, defined to be the ratio between OOD accuracy and natural accuracy. MNIST-C includes 15 carefully chosen corruptions, covering a broad range of corruptions that are not characterized by adversarial robustness while preserving the semantics. We evaluate models trained with both adversarial training and certified training under $\epsilon = 0.1$ and $\epsilon = 0.3$, and report the corresponding OOD accuracy of the model trained via standard training. We note that none of the models has seen these corruptions during training.

Figure 6 depicts the result of OOD generalization for each model on all corruptions. We observe the following: (1) certified training improves OOD generalization compared to standard training except on the *brightness* corruption where both adversarial and certified training fails; (2) certified training shows different OOD generalization patterns to adversarial training, *e.g.*, certified training boost generalization on the *canny edges* corruption while adversarial training wins on the *stripe* corruption. In general, we find that certified training either greatly boosts the OOD generalization or significantly downgrades the OOD generalization depending on the corruption, and the bad cases are usually those in which adversarial training performs worse than or similarly to standard training. Therefore, we hypothesize that these corruptions are at odds with adversarial robustness. Further, different training $\epsilon$ does not significantly affect the OOD generalization except few cases, and ranking in certified accuracy does not show strong relations with the ranking in OOD generalization. Overall, these results suggest that certified training has the potential to improve OOD generalization to corruptions that standard training struggles with, and the effect is exaggerated when adversarial training improves over standard training. More results on CIFAR-10-C (Hendrycks & Dietterich, 2019) are included in App. C as Figure 9.

## 6 Future Directions

We now summarize directions for future improvements of certified training and its potential applications. As shown in Section 5.2, certified models make shared mistakes on some hard samples, thus curriculum learning with some well-defined difficulty ranking could facilitate training, where optimization has been known to be particularly hard (Jovanović et al., 2022). Moreover, in Section 5.3 we showed that even the most trainable method, IBP, struggles to keep necessary model utilization on large $\epsilon$. Therefore, future work is still required to improve the learning process of certified training. Despite the challenges, in Section 5.5 we find that certified models can have surprising and qualitatively different behavior on OOD generalization, which could be a promising application for certified training beyond certified robustness.

## 7 Conclusion

We introduced CTBench, a unified library and high-quality benchmark for deterministic certified training on $L_\infty$ robustness. Based on CTBench, we extensively evaluated certified models trained via state-of-the-art methods, analyzing their regularization strength and utilities. Our analysis reveals that certified training schemes can reduce loss fragmentation, adaptively keep model utilization, make shared mistakes, and generalize well on data with certain corruptions. We are confident that the insights and tools provided by CTBench will facilitate future research on certified training and its applications.

## Reproducibility Statement

We release the complete codebase of CTBench, including the implementation of all certified training methods and the model checkpoints for the benchmark. The codebase is available at ANONYMIZED (available in the supplementary material). A complete description of the experiment setup and hyperparameters is provided in App. B.

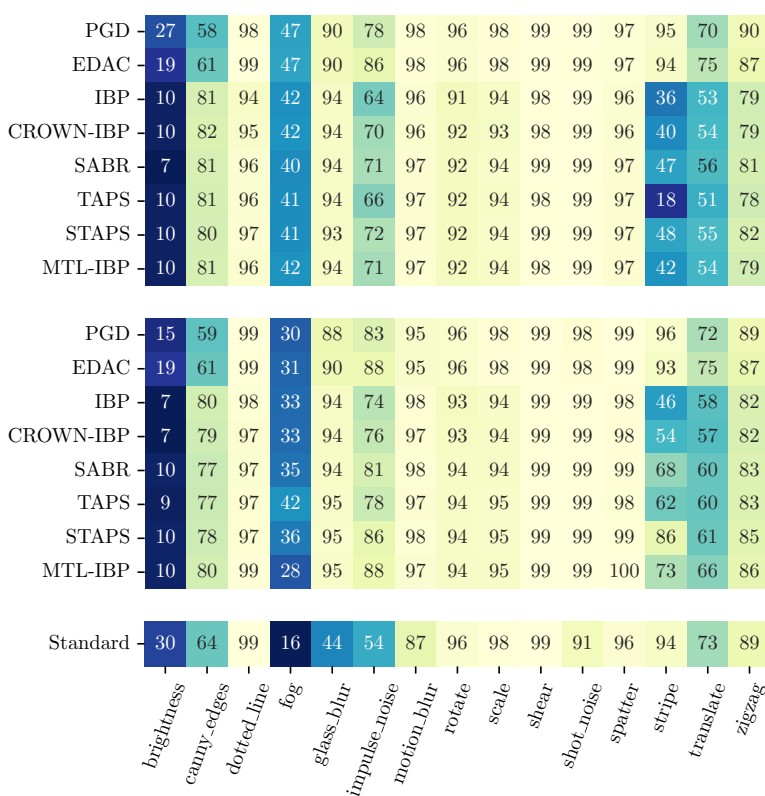

Figure 6: Out-of-distribution generalization evaluated on MNIST-C for models trained on MNIST at $\epsilon = 0.3$ (top), $\epsilon = 0.1$ (middle) and standard training (bottom).

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

# A DISCUSSION

## A.1 DECOMPOSITION OF IMPROVEMENTS

Decomposition of the universal modifications we made such as batch norm fixes and the hyperparameter tuning is not always possible, as these modifications allow additional vectors of hyperparameter for tuning. For example, we fix batch norm statistics in one batch rather than reset it multiple times as done in some original implementations, allowing $w_{rob}$ to be tuned within $[0, 1]$, while in the literature $w_{rob}$ has to be fixed to 1. Therefore, we cannot formally decompose the effects of hyperparameter tuning and batch norm behaviors, as they are closely dependent on each other.

The PreciseBN (Wu & Johnson, 2021) that we adopt, which is to set batch norm statistics based on the entire training dataset at test time, does not change the training at all, since at every training step batch norm layers are set by batch statistics. Therefore, this only smooths the test time performance and potentially improves the final performance. While this is good for monitoring the learning curve, the final performance improvement is minimal in our experiments, and in most cases almost no improvement on the final model is observed. This is expected since batch norm statistics also converge when the model converges.

The literature results are run with three different random seeds, and only the best results among them are reported. This prevents us from substituting our fine-tuned hyperparameter to the original implementation because merely using the same hyperparameters even based on the original implementation hardly reproduces the same number as reported in the literature. In contrast, we run every experiment with the same fixed random seed to allow fair and faithful comparison. Nevertheless, we can showcase the effect for one setting: IBP on MNIST $\epsilon = 0.3$. The literature reports 93.1% certified accuracy, while the same hyperparameter results in 93.18% in our implementation. Further tuning the hyperparameters as in the CTBench benchmark gets 93.8%. While this proves the effectiveness of both the implementation and our hyperparameter tuning, we would like to note that based on previous arguments, this does not faithfully decompose the effect of hyperparameter tuning and batch norm changes, and such decomposition efforts are doomed to fail.

In summary, while decomposition is beneficial, there are practical concerns preventing us from formally decomposing the effects. However, since this work introduces a library and benchmark rather than precisely decomposing the effect of each beneficial change, this does not undermine the contribution of this work.

## A.2 LIMITATIONS

The main limitation of CTBENCH is that we only consider deterministic certified training, while randomized certified robustness (Cohen et al., 2019) has also made substantial progress. Moreover, we only consider the adversarial robustness, while other types of robustness, such as robustness against patch attacks (Salman et al., 2022) is also important. Finally, we only focus on $L_\infty$ robustness, and leave the discussion about other norms as future work.

## A.3 BROADER IMPACTS

This work focuses on certified defenses against adversarial attacks, which is a crucial component of trustworthy artificial intelligence. The proposed benchmark CTBENCH will facilitate future research on certified training and its applications. The insights and tools provided by CTBENCH will help researchers to develop more robust and reliable machine learning models. The potential harm of this work are as follows:

- Certified models can provide a fake security when the models are applied against non-adversarial perturbations.

- Certification methods are computationally expensive, which will consume more energy and thus possibly harm the environment.

# B EXPERIMENT DETAILS

## B.1 DATASET

We use the MNIST (LeCun et al., 2010), CIFAR-10 (Krizhevsky et al., 2009) and TINYIMAGENET (Le & Yang, 2015) datasets for our experiments. All are open-source and freely available with unspecified license. The data preprocessing mostly follows De Palma et al. (2024). For MNIST, we do not apply any preprocessing. For CIFAR-10 and TINYIMAGENET, we normalize with the dataset mean and standard deviation and augment with random horizontal flips. We apply random cropping to $32 \times 32$ after applying a 2 pixel zero padding at every margin for CIFAR-10, and random cropping to $64 \times 64$ after applying a 4 pixel zero padding at every margin for TINYIMAGENET. We train on the corresponding train set and certify on the validation set, as adopted in the literature (Shi et al., 2021; Müller et al., 2023; Mao et al., 2023; De Palma et al., 2024).

## B.2 MODEL ARCHITECTURES

We follow Shi et al. (2021); Müller et al. (2023) and use a `CNN7` with Batch Norm for our main experiments. `CNN7` is a convolutional network with 7 convolutional and linear layers. All but the last linear layer are followed by a Batch Norm and ReLU layer. This architecture is found to achieve uniformly better results across settings (Shi et al., 2021), and thus is adopted by the literature (Shi et al., 2021; Müller et al., 2023; Mao et al., 2023; De Palma et al., 2024). For TINYIMAGENET, the stride of the last convolution is doubled to reduce the cost.

## B.3 TRAINING DETAILS

**Initialization** Adversarial training methods are initialized by Kaiming uniform (He et al., 2015), while certified training methods are initialized by IBP initialization (Shi et al., 2021).

**Training Schedule** We mostly follow the training schedule of (De Palma et al., 2024), but in some cases a shorter schedule to reduce cost. Specifically, the warmup phase is 20 epochs for MNIST $\epsilon = 0.1$ and $\epsilon = 0.3$, 80 epochs for CIFAR-10 $\epsilon = \frac{2}{255}$, 120 epochs for CIFAR-10 $\epsilon = \frac{8}{255}$ and 80 epochs for TINYIMAGENET $\epsilon = \frac{1}{255}$. In addition, for CIFAR-10 and TINYIMAGENET, we use standard training for 1 additional epoch at the beginning. We apply the IBP regularization proposed by (Shi et al., 2021), with weight equals 0.5 on MNIST and CIFAR-10, and 0.2 on TINYIMAGENET, during the warmup phase. In total, we train 70 epochs for MNIST $\epsilon = 0.1$ and $\epsilon = 0.3$, 160 epochs for CIFAR-10 $\epsilon = \frac{2}{255}$, 240 epochs for CIFAR-10 $\epsilon = \frac{8}{255}$, and 160 epochs for TINYIMAGENET $\epsilon = \frac{1}{255}$.

**Optimization** We use Adam (Kingma & Ba, 2015) with a learning rate of 0.0005. The learning rate is decayed by a factor of 0.2 at epoch 50 and 60 for MNIST $\epsilon = 0.1$ and $\epsilon = 0.3$, at epoch 120 and 140 for CIFAR-10 $\epsilon = \frac{2}{255}$, at epoch 200 and 220 for CIFAR-10 $\epsilon = \frac{8}{255}$, and at epoch 120 and 140 for TINYIMAGENET $\epsilon = \frac{1}{255}$. We use a batch size of 256 for MNIST, and 128 for CIFAR-10 and TINYIMAGENET. Gradients of each step are clipped to 10 in $L_2$ norm. No weight decay is applied and $L_1$ regularization only on weights of linear and convolution layers is used.

## B.4 TUNING SCHEME

We conduct a hyperparameter tuning for each method to ensure the best performance, and reduce the search space whenever appropriate based on human knowledge. The search space for each hyperparameter is as follows:

- $L_1$ *regularization*: $\{1 \times 10^{-6}, 2 \times 10^{-6}, 5 \times 10^{-6}, 1 \times 10^{-5}, 2 \times 10^{-5}, 5 \times 10^{-5}\}$. We include $3 \times 10^{-6}$ specifically for CIFAR-10 $\epsilon = \frac{2}{255}$, as this is the value reported by De Palma et al. (2024).

- $w_{rob}$: $\{0.7, 0.8, 0.9, 1.0\}$. Surprisingly, $w_{rob}$ not equal to 1 can improve both certified and natural accuracy by a large margin when $\epsilon$ is small.

- *Train* $\epsilon$: we use 2x train $\epsilon$ for MNIST $\epsilon = 0.1$, and tune within $\{1x, 1.25x, 1.5x\}$ specifically for CIFAR-10 $\epsilon = \frac{2}{255}$. For others, we use the test $\epsilon$ for training.

Table 3: Best hyperparameter for MNIST $\epsilon = 0.1$.

|  | PGD | EDAC | IBP | CROWN-IBP | SABR | TAPS | STAPS | MTL-IBP |
|---|---|---|---|---|---|---|---|---|
| $L_1$ regularization | $1 \times 10^{-5}$ | $1 \times 10^{-5}$ | $2 \times 10^{-6}$ | $2 \times 10^{-6}$ | $1 \times 10^{-6}$ | $1 \times 10^{-6}$ | $1 \times 10^{-6}$ | $1 \times 10^{-5}$ |
| $w_{\text{rob}}$ | 1.0 | 1.0 | 1.0 | 1.0 | 0.7 | 0.7 | 0.7 | 0.7 |
| Train $\epsilon$ | 0.2 | 0.2 | 0.2 | 0.2 | 0.2 | 0.2 | 0.2 | 0.2 |
| $\epsilon$ shrink ratio | / | / | / | / | 0.4 | / | 0.4 | / |
| Classifier size | / | / | / | / | / | 3 | 1 | / |
| TAPS gradient scale | / | / | / | / | / | 4 | 4 | / |
| ReLU shrink ratio | / | / | / | / | / | / | / | / |
| IBP coefficient | / | / | / | / | / | / | / | 0.02 |

Table 4: Best hyperparameter for MNIST $\epsilon = 0.3$.

|  | PGD | EDAC | IBP | CROWN-IBP | SABR | TAPS | STAPS | MTL-IBP |
|---|---|---|---|---|---|---|---|---|
| $L_1$ regularization | $5 \times 10^{-6}$ | $5 \times 10^{-6}$ | $1 \times 10^{-6}$ | $1 \times 10^{-6}$ | $2 \times 10^{-6}$ | $2 \times 10^{-6}$ | $2 \times 10^{-6}$ | $1 \times 10^{-6}$ |
| $w_{\text{rob}}$ | 1.0 | 1.0 | 1.0 | 1.0 | 1.0 | 1.0 | 1.0 | 1.0 |
| Train $\epsilon$ | 0.3 | 0.3 | 0.3 | 0.3 | 0.3 | 0.3 | 0.3 | 0.3 |
| $\epsilon$ shrink ratio | / | / | / | / | 0.8 | / | 0.8 | / |
| Classifier size | / | / | / | / | / | 1 | 1 | / |
| TAPS gradient scale | / | / | / | / | / | 3 | 1 | / |
| ReLU shrink ratio | / | / | / | / | / | / | / | / |
| IBP coefficient | / | / | / | / | / | / | / | 0.5 |

- $\epsilon$ *shrink ratio for* SABR *and* STAPS: we mostly keep the value in the literature. When we observe large certifibility gap, we increase the shrink ratio by $0.1$ until the performance fails to increase consistently.

- *Classifier size for* TAPS *and* STAPS: we keep the value in the literature for TAPS, and include only 1 ReLU layer in the classifier for STAPS universally.

- TAPS *gradient scale*: $\{1, 2, 3, 4, 6, 8\}$.

- *ReLU shrink ratio for* SABR *and* STAPS: we keep the value in the literature, thus shrinking the output box of each ReLU by multiplying $0.8$ on CIFAR-10 $\epsilon = \frac{2}{255}$ and do not apply this in other settings.

- IBP *coefficient for* MTL-IBP: $\{0.01, 0.02, 0.05\}$ for MNIST $\epsilon = 0.1$, CIFAR-10 $\epsilon = \frac{2}{255}$ and TINYIMAGENET $\epsilon = \frac{1}{255}$, and $\{0.4, 0.5, 0.6\}$ for MNIST $\epsilon = 0.3$, CIFAR-10 $\epsilon = \frac{8}{255}$.

- *Attack Strength*: we use 3 restarts everywhere for the attack. By default, we use 10 steps for MNIST $\epsilon = 0.1$, 5 steps for MNIST $\epsilon = 0.3$, 8 steps for CIFAR-10 $\epsilon = \frac{2}{255}$, 10 steps for CIFAR-10 $\epsilon = \frac{8}{255}$, and 1 step for TINYIMAGENET $\epsilon = \frac{1}{255}$. However, we find MTL-IBP benefits from using only 1 step everywhere, while more steps will hurt certified accuracy, thus we only use 1 step specifically for MTL-IBP except CIFAR-10 $\epsilon = \frac{2}{255}$, consistent to De Palma et al. (2024). We further only use 2x attack $\epsilon$ for MTL-IBP on CIFAR-10 $\epsilon = \frac{2}{255}$.

We report the best hyperparameter for each method respectively in Table 3, Table 4, Table 5, Table 6, and Table 7.

Table 5: Best hyperparameter for CIFAR-10 $\epsilon = 2/255$.

|  | PGD | EDAC | IBP | CROWN-IBP | SABR | TAPS | STAPS | MTL-IBP |
|---|---|---|---|---|---|---|---|---|
| $L_1$ regularization | $2 \times 10^{-5}$ | $5 \times 10^{-6}$ | $1 \times 10^{-6}$ | $1 \times 10^{-6}$ | $1 \times 10^{-6}$ | $2 \times 10^{-6}$ | $5 \times 10^{-6}$ | $3 \times 10^{-6}$ |
| $w_{\text{rob}}$ | 1.0 | 1.0 | 1.0 | 1.0 | 0.7 | 1.0 | 1.0 | 0.9 |
| Train $\epsilon$ | 2/255 | 2/255 | 2/255 | 2/255 | 3/255 | 2/255 | 3/255 | 2/255 |
| $\epsilon$ shrink ratio | / | / | / | / | 0.1 | / | 0.1 | / |
| Classifier size | / | / | / | / | / | 5 | 1 | / |
| TAPS gradient scale | / | / | / | / | / | 5 | 5 | / |
| ReLU shrink ratio | / | / | / | / | 0.8 | / | 0.8 | / |
| IBP coefficient | / | / | / | / | / | / | / | 0.01 |

Table 6: Best hyperparameter for CIFAR-10 $\epsilon = 8/255$.

|  | PGD | EDAC | IBP | CROWN-IBP | SABR | TAPS | STAPS | MTL-IBP |
|---|---|---|---|---|---|---|---|---|
| $L_1$ regularization | $1 \times 10^{-6}$ | $1 \times 10^{-6}$ | 0 | 0 | 0 | 0 | 0 | 0 |
| $w_{\text{rob}}$ | 1.0 | 1.0 | 1.0 | 1.0 | 1.0 | 1.0 | 1.0 | 1.0 |
| Train $\epsilon$ | 8/255 | 8/255 | 8/255 | 8/255 | 8/255 | 8/255 | 8/255 | 8/255 |
| $\epsilon$ shrink ratio | / | / | / | / | 0.7 | / | 0.9 | / |
| Classifier size | / | / | / | / | / | 1 | 1 | / |
| TAPS gradient scale | / | / | / | / | / | 2 | 2 | / |
| ReLU shrink ratio | / | / | / | / | / | / | / | / |
| IBP coefficient | / | / | / | / | / | / | / | 0.5 |

Table 7: Best hyperparameter for TINYIMAGENET $\epsilon = 1/255$.

|  | PGD | EDAC | IBP | CROWN-IBP | SABR | TAPS | STAPS | MTL-IBP |
|---|---|---|---|---|---|---|---|---|
| $L_1$ regularization | $5 \times 10^{-5}$ | $1 \times 10^{-5}$ | $1 \times 10^{-5}$ | $1 \times 10^{-5}$ | $1 \times 10^{-5}$ | $1 \times 10^{-5}$ | $1 \times 10^{-5}$ | $5 \times 10^{-5}$ |
| $w_{\text{rob}}$ | 1.0 | 1.0 | 1.0 | 1.0 | 1.0 | 1.0 | 1.0 | 0.7 |
| Train $\epsilon$ | 1/255 | 1/255 | 1/255 | 1/255 | 1/255 | 1/255 | 1/255 | 1/255 |
| $\epsilon$ shrink ratio | / | / | / | / | 0.4 | / | 0.6 | / |
| Classifier size | / | / | / | / | / | 1 | 1 | / |
| TAPS gradient scale | / | / | / | / | / | 8 | 4 | / |
| ReLU shrink ratio | / | / | / | / | / | / | / | / |
| IBP coefficient | / | / | / | / | / | / | / | 0.05 |

## B.5 CERTIFICATION DETAILS

We combine IBP (Gowal et al., 2018), CROWN-IBP (Zhang et al., 2020), and MN-BAB (Ferrari et al., 2022) for certification running the most precise but also computationally costly MN-BAB only on samples not certified by the other methods. The timout for each input is set to 1000 seconds.

## B.6 COMPUTATION

We train and certify MNIST $\epsilon = 0.1$, MNIST $\epsilon = 0.3$ and CIFAR-10 $\epsilon = \frac{8}{255}$ models on a single NVIDIA GeForce RTX 2080 Ti with Intel(R) Xeon(R) Silver 4214R CPU @ 2.40GHz and 530GB RAM. We train and certify CIFAR-10 $\epsilon = \frac{2}{255}$ and TINYIMAGENET $\epsilon = \frac{1}{255}$ models on a single NVIDIA L4 with Intel(R) Xeon(R) CPU @ 2.20GHz CPU and 377 GB RAM. The training and certification time for each method is reported in Table 8.

## C ADDITIONAL RESULTS

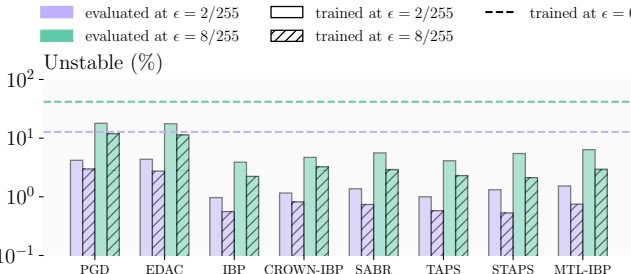

Figure 7: Ratio of unstable neurons for models trained on CIFAR-10 with different methods and $\epsilon$.

Table 8: Training and certification time for each method on different datasets and $\epsilon$.

| Dataset | $\epsilon$ | Method | Train Time (seconds) | Certification Time (seconds) |
|---|---|---|---|---|
| MNIST | 0.1 | PGD | $1.5 \times 10^4$ | / |
| | | EDAC | $3.1 \times 10^4$ | / |
| | | IBP | $2.1 \times 10^3$ | $2.5 \times 10^3$ |
| | | CROWN-IBP | $5.6 \times 10^3$ | $1.8 \times 10^3$ |
| | | SABR | $1.8 \times 10^4$ | $6.0 \times 10^3$ |
| | | TAPS | $3.8 \times 10^4$ | $6.0 \times 10^3$ |
| | | STAPS | $2.5 \times 10^4$ | $6.9 \times 10^3$ |
| | | MTL-IBP | $6.8 \times 10^3$ | $6.8 \times 10^3$ |
| | 0.3 | PGD | $1.1 \times 10^4$ | / |
| | | EDAC | $2.2 \times 10^4$ | / |
| | | IBP | $2.6 \times 10^3$ | $3.2 \times 10^4$ |
| | | CROWN-IBP | $5.4 \times 10^3$ | $2.6 \times 10^4$ |
| | | SABR | $9.7 \times 10^3$ | $5.2 \times 10^4$ |
| | | TAPS | $7.1 \times 10^3$ | $4.7 \times 10^4$ |
| | | STAPS | $1.4 \times 10^4$ | $5.1 \times 10^4$ |
| | | MTL-IBP | $5.5 \times 10^3$ | $4.4 \times 10^4$ |
| CIFAR-10 | $\frac{2}{255}$ | PGD | $2.8 \times 10^4$ | / |
| | | EDAC | $1.3 \times 10^5$ | / |
| | | IBP | $1.2 \times 10^4$ | $1.3 \times 10^5$ |
| | | CROWN-IBP | $2.7 \times 10^4$ | $1.9 \times 10^5$ |
| | | SABR | $2.4 \times 10^4$ | $1.6 \times 10^5$ |
| | | TAPS | $1.1 \times 10^5$ | $1.1 \times 10^5$ |
| | | STAPS | $4.5 \times 10^4$ | $3.0 \times 10^5$ |
| | | MTL-IBP | $3.6 \times 10^4$ | $2.7 \times 10^5$ |
| | $\frac{8}{255}$ | PGD | $6.4 \times 10^4$ | / |
| | | EDAC | $1.3 \times 10^5$ | / |
| | | IBP | $1.1 \times 10^4$ | $1.9 \times 10^4$ |
| | | CROWN-IBP | $2.1 \times 10^4$ | $2.0 \times 10^4$ |
| | | SABR | $4.1 \times 10^4$ | $6.5 \times 10^4$ |
| | | TAPS | $3.3 \times 10^4$ | $4.0 \times 10^4$ |
| | | STAPS | $9.9 \times 10^4$ | $4.2 \times 10^4$ |
| | | MTL-IBP | $2.2 \times 10^4$ | $5.6 \times 10^4$ |
| TINYIMAGENET | $\frac{1}{255}$ | PGD | $1.0 \times 10^5$ | / |
| | | EDAC | $2.0 \times 10^5$ | / |
| | | IBP | $6.7 \times 10^4$ | $4.9 \times 10^3$ |
| | | CROWN-IBP | $2.0 \times 10^5$ | $1.3 \times 10^4$ |
| | | SABR | $1.1 \times 10^5$ | $1.8 \times 10^4$ |
| | | TAPS | $2.8 \times 10^5$ | $1.5 \times 10^4$ |
| | | STAPS | $3.3 \times 10^5$ | $2.6 \times 10^4$ |
| | | MTL-IBP | $1.5 \times 10^5$ | $5.1 \times 10^3$ |

Table 9: Observed count of common mistakes of certification algorithms (MN-BAB (Ferrari et al., 2022) and OVAL (De Palma et al., 2022)) on MNIST against their expected values assuming independence across certification mistakes.

| | | neither certify | one certifies | both certify |
|---|---|---|---|---|
| $\epsilon = 2/255$ | obs. | 3549 | 15 | 6436 |
| | exp. | 1264 | 4585 | 4151 |
| $\epsilon = 8/255$ | obs. | 6454 | 9 | 3537 |
| | exp. | 4171 | 4575 | 1254 |

Table 10: Observed count of common mistakes on CIFAR-10 against their expected values assuming independence across model mistakes.

| | | # models succeeded | | | | | | |
|---|---|---|---|---|---|---|---|---|
| | | 0 | 1 | 2 | 3 | 4 | 5 | 6 |
| $\epsilon = \frac{2}{255}$ | obs. | 2350 | 653 | 520 | 564 | 708 | 894 | 4311 |
| | exp. | 35 | 330 | 1296 | 2704 | 3163 | 1965 | 507 |
| $\epsilon = \frac{8}{255}$ | obs. | 5206 | 679 | 487 | 388 | 387 | 585 | 2268 |
| | exp. | 766 | 2457 | 3283 | 2339 | 937 | 200 | 18 |

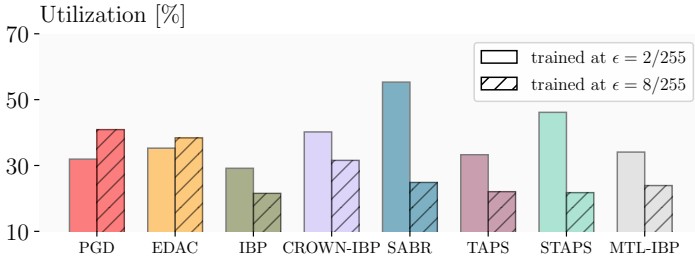

Figure 8: Model utilization for models trained on CIFAR-10 with different methods and $\epsilon$. We note that standard training has 35.79% utilization.

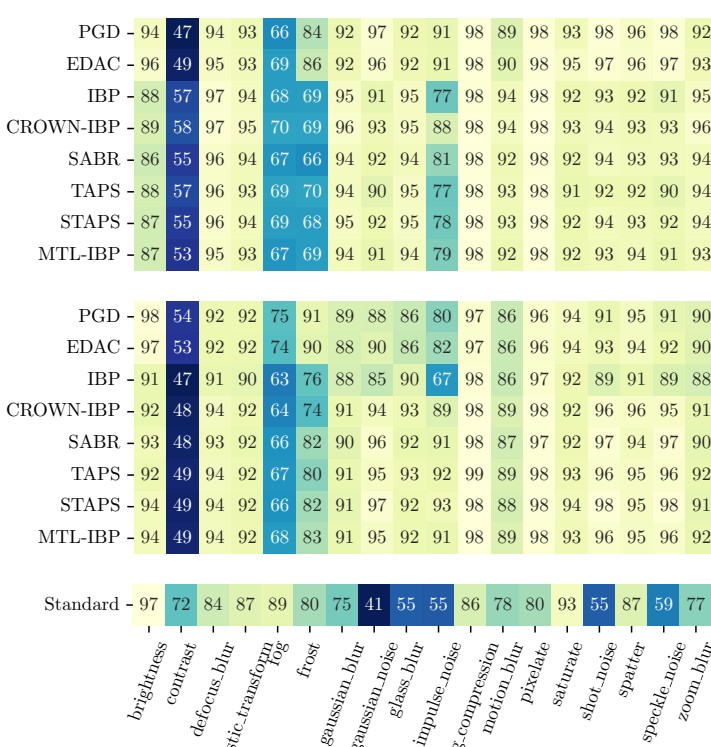

Figure 9: Out-of-distribution generalization evaluated on CIFAR-10-C for models trained on CIFAR-10 at $\epsilon = 8/255$ (top), $\epsilon = 2/255$ (middle) and standard training (bottom).

