# OpenReview forum: "CTBench: A Library and Benchmark for Certified Training"
_ICLR.cc/2025/Conference — Submitted to ICLR 2025_

### Official Review · Reviewer_s9m2 · 2024-10-22

**Soundness:** 4
**Presentation:** 4
**Contribution:** 2
**Rating:** 5
**Confidence:** 4

**Summary:**

The The paper introduces CTBENCH, a unified library and benchmark for evaluating certified training methods for neural networks. It addresses the challenges in comparing existing certified training algorithms by standardizing training schedules, certification methods, and hyperparameter tuning. The authors demonstrate that most algorithms in CTBENCH surpass previously reported results, revealing that much of the perceived advantage of newer methods diminishes when outdated baselines are properly tuned. The benchmark provides insights into certified training methods, encouraging future research by offering a consistent framework and re-establishing state-of-the-art performance.

**Strengths:**

1. The paper is well presented, well written, with clear goals and objectives.

2- While this is for a non-expert is not obvious, but the amount of experiments and computation required in this paper is beyond impressive.

3- The insights of the paper are particularly helpful. I personally did not expect that current SOTA methods are under performing. However, it was not that surprising that the improvements over IBP for larger epsilons are not that big.

4- Paper sheds light on a relatively good problem.

**Weaknesses:**

1. The paper focuses solely on deterministic certified training, overlooking advancements in randomized certified robustness. I believe the paper should have cited works like Cohen et al., Matthias et al. (l1 certification with differential privacy -- early works from 2019), Greg Yang ("All Shapes and Sizes" paper), among many others.

2. The paper only considers infinity ball, neglecting other perturbation sets. While this is generally okay, some insights with a few experiments in other perturbation sets might be helpful. It is not clear whether the proposed tricks in the library as part of the unified certified training would work for other perturbation sets (e.g., L2). If they do not, it raises the question of whether we would need a separate library for each perturbation set. The next steps are unclear if that is the case.

3. Some conclusions on the impact of tuning and modifications, while valid, lack formal decomposition, making it difficult to quantify individual contributions. No clarity on the contribution of each individual component (batch norm, etc) towards the final performance. A small systematic study will be very helpful.

4. The evaluation is based on a single model architecture (CNN7); the paper should demonstrate that the library and recommendations hold across different architectures.


General comment: Interest in certified models has significantly declined over the past two years. At ECCV, for example, there were notably fewer submissions and accepted papers on adversarial attacks, even though this topic was previously very popular in vision conferences. One reason for this decline could be the uncertainty around where such certifications can be practically deployed, especially given the massive scale of current models, which are thousands of times larger than the CNNs discussed here. Furthermore, as models shift towards generative architectures, it’s unclear who will find this domain relevant. While the paper makes valuable contributions, this direction feels somewhat outdated by about two years and the question of the benefit for it is very unclear and vague, at least to me. I would love to hear the authors take on this.

Minor Comments:
1. Cite "is NP-complete" line 321.
2.  Is not typical robust accuracy (adv acc) for PGD around 48% on 8/255 CIFAR10? Or is because you use CNN7.
3. adversarial accuracy is not well defined in line 135. You need to say that it is empirical and serves as an upper bound to the robust accuracy.
4. certified accuracy defined in 133 is not correct. It should be the portion of *correctly* classified samples that are certifiably robust.

**Questions:**

See above; I would love to hear the authors comments on each of the weakness above along with a response to the general comment.

---

> ### Author Response · Authors · 2024-11-19
> **Response to $\Rs$**
>
> We are happy to hear that Reviewer $\Rs$ considers our work is well-written, utilizes “beyond impressive” amounts of experiments and computational resources on a good problem, and derives particularly helpful insights into the deterministic certified training.
>
> **Q1: Should this paper overlook the advancements in randomized certified robustness?**
>
> As clearly stated at the beginning of the abstract and introduction, we focus solely on deterministic certified robustness. We acknowledge the advances in randomized certified robustness, but randomized certificates are not comparable to deterministic certificates. For example, one provides high confidence certificates while the other provides deterministic certificates, and one brings additional inference overhead while the other does not have inference overhead. Therefore, as an in-depth study regarding deterministic certified robustness, we do not refer to randomized smoothing literature.
>
> **Q2: This paper only considers $L_\infty$ norm; could the authors provide some insights into other perturbation norms? Do we need a separate library in case one wants to implement other norms?**
>
> Unfortunately, the field of deterministic certified radius focuses on the $L_\infty$ norm and no deterministic certified training algorithms for other norms have been developed. This makes us unable to provide any insights into other perturbation norms. However, we choose to design our library such that norm types are disentangled: if one wants to implement another norm, they can easily migrate their solution into our library, as relaxations (core aspects about norms) are modularized and thus extensible. This allows development regarding other norms in the future.
>
> **Q3: Could you decompose the improvements and quantify contributions of individual components?**
>
> Great question! We discuss improvement decomposition thoroughly in Appendix A.1. In short, improvements often bring additional hyperparameter vectors for tuning, and thus formal decomposition of benefits is not practically feasible. More details can be found in Appendix A.1.
>
> **Q4: Is the developed library extensible to other architectures other than the state-of-the-art CNN7?**
>
> Yes, architectures are modularized, and thus incorporating another architecture is trivial for our library. In particular, the current library has multiple architectures implemented; we report CNN7 in the paper because this is the state-of-the-art architecture and represents the most important aspects of certified training.
>
> **Q5: Interest in (adversarially) robust models seem to have declined recently; potential reasons include practicability and the increased interest in generative models. While this paper makes valuable contributions (towards certified robustness), is this direction outdated and no longer meaningful?**
>
> We are glad to discuss this from our own perspective, but not representing ideas in general. Adversarial robustness is an essential requirement for artificial intelligence, thus it will never be outdated or meaningless until we solve it. In addition, adversarial robustness is not losing interest, as many works are shifted to jailbreaking or discovering other attack vectors for generative models. Furthermore, many start-ups regarding model robustness have been established, thus its practicability in certain areas has been acknowledged. Many legitimate rules are also developed, and one frequent requirement is robustness. Therefore, we believe the decreased number of publications does not mean declined necessity; instead, it represents a hard time for problem solvers, because this problem has been shown to be non-trivial.
>
> **Q6: Is adversarial robustness for PGD lower than the state-of-the-art adversarial robustness? Is this because the authors use CNN7 (rather than a much larger model used in the adversarial machine learning literature)?**
>
> Yes, this is because we use CNN7, for a fair comparison with certified models.
>
> **Q7: Does L133 define certified accuracy imprecisely?**
>
> Yes, thanks for the correction!

---

> > ### Comment · Reviewer_s9m2 · 2024-11-25
> > **Follow up**
> >
> > I want to first take the opportunity to thank the authors for their efforts in the paper and rebuttals. I have read all the reviews and the authors' responses to each review.
> >
> > Q1: I generally disagree. This is a question of whether one would prefer a high-confidence (where the confidence level is controlled) probabilistic certificate that scales to hundreds of layers versus a deterministic certificate that scales to tens of layers. The answer strongly depends on the application. However, my disagreement with the authors here is a matter of subjective taste and does not influence my final decision. I include it here for completeness only.
> >
> > Q2: Point taken. I do not mind 𝐿∞ balls, or any ball for that matter, as they are all equivalent in some metric space up to a constant.
> >
> > Q3: I thank the authors for their feedback. However, this presents a scalability issue. If every method requires tuning these parameters, let alone introducing new ones, the general utility of a library becomes questionable. What is the value of such a baseline if, for every new method, we must revisit and fine-tune it with the proposed set of parameters?
> >
> > Q4: Addressed.
> >
> > Q5: I disagree here again. Hijacking, prompting, and similar issues are not related to certification but rather to empirical robustness, which has gained traction in vision and is now being explored in the language domain. We have not solved certification for vision, let alone for language, which involves challenges ranging from model scale to discrete optimization over tokens. The majority of startups I know of in this space focus on empirical evaluation layers, red-teaming, and jailbreak prevention, with very few (if any) claiming provable guarantees against generation. If a company were to achieve this, it would indeed be a significant breakthrough, as this is an open problem. I do not believe we yet have the algorithms to accomplish this—scaling deterministic methods to hundreds of layers in vision alone has proven challenging. The decline in research papers on this topic is not definitive proof but rather evidence of waning interest. That said, my rating is not based solely on Q5.
> >
> > Q6: Addressed.
> >
> > Q7: Addressed.
> >
> >
> >  thank the authors again for their efforts on the paper and rebuttal. I am keeping my score unchanged, but I will not oppose if other reviewers feel differently. The reasons behind my score are summarized as follows: (1) Certification, while important, has not yet demonstrated scalability. We are still reporting results on CNN7, a model from several years ago, with only a few layers. Scale remains a major issue, raising questions about the value of this benchmark. (2) The library requires fine-tuning several parameters for new methods, which limits the core utility of the proposed approach. (3) The combination of points (1) and (2) amplifies my concerns about the applicability and usability of the work. The need to tune hyperparameters for the library to benchmark new models each time, even if the models are small, highlights the lack of scalability in certification.

---

> > > ### Author Response · Authors · 2024-11-25
> > > **Reply to Reviewer $\Rs$**
> > >
> > > We thank Reviewer $\Rs$ for their response and are happy to know that we have addressed all technical concerns from the reviewer. In the following, we provide further clarification regarding the reviewer's remaining concerns, including factual corrections and our personal perspectives.
> > >
> > > **On the Comparison Between Deterministic and Randomized Certification**
> > >
> > > We understand the reviewer's perspective and appreciate their acknowledgment that this distinction is largely a matter of application preferences. However, we would like to reiterate that our claim regarding the non-comparability of deterministic and randomized certificates is grounded in fundamental differences between the two approaches. Deterministic certificates guarantee robustness with absolute certainty, while randomized methods provide probabilistic guarantees dependent on sampling and confidence levels, also incurring a multiplicative computational overhead factor at inference time. These differences are not just subjective preferences but intrinsic properties of the methods, each suited to specific needs. In particular, these differences lead to major technical deviations in the methodology between deterministic and randomized certified robustness.
> > >
> > > **Scalability and Hyperparameter Tuning**
> > >
> > > We regret any confusion about the necessity of hyperparameter tuning. To clarify, the tuning effort that the reviewer referred to applies to benchmarking new methods, not the library itself. The library is designed to be general and extensible, making it a robust foundation for future research. Benchmarking inherently requires computational and optimization effort, but this is a natural part of scientific evaluation rather than a limitation of the library.
> > >
> > > We understand the concern about scalability and agree that certification benchmarks often involve significant computational resources. However, this effort is crucial for progress, as robust and fair comparisons require careful evaluation. When designing new methods, tuning both general-interest hyperparameters (e.g. the level of L1 regularization) and method-specific ones (e.g. $\lambda$ in SABR [2] or $\alpha$ in MTL-IBP [4]) is unavoidable, but we hope that our library will help reduce the overall time and effort spent on these experiments by providing a modular and extensible framework to ease the burden of implementation and testing.
> > >
> > > **On the State of the Field**
> > >
> > > We thank the reviewer for sharing their views on the broader challenges in certified robustness. We now realize that the reviewer wes referring specifically to certified robustness in this context, rather than general adversarial robustness. Certified robustness offers critical benefits, such as verifiable guarantees of model behavior under specific perturbations, which empirical robustness methods cannot provide.
> > >
> > > While we acknowledge the difficulties in scaling certified methods, recent advancements (published in top-tier conferences in recent years) [1-6] demonstrate that the field continues to progress. These works explore novel algorithms, larger architectures, and improved training paradigms to address scalability and robustness challenges. The reviewer might refer to the diminished interest in  certification algorithms which indeed has fewer publications due to the ability and completeness of existing certification methods, but the interest apparently has shifted to certified training which trains/designs the network such that they are easier to certify. Certified training, therefore, is the main focus of our work.
> > >
> > > We remain optimistic about the general certified robustness field despite its difficulties. Hard problems like these demand persistence, as breakthroughs often emerge from cumulative effort over time. The current challenges highlight the need for innovative solutions, which motivates our work and contributions.
> > >
> > > **Final Comment**
> > >
> > > While we understand Reviewer $\Rs$’s concerns about scalability and usability, we believe our work contributes valuable tools and insights that pave the way for addressing these very challenges.
> > >
> > > We share Reviewer’s optimism that certified robustness, while difficult, remains a meaningful and necessary pursuit. Without tackling such hard problems, progress in ensuring robust and trustworthy AI systems would stall.
> > >
> > > We thank Reviewer $\Rs$ again for their feedback, and we are grateful for their willingness to engage deeply with our work.
> > >
> > > **References**
> > >
> > > [1] Shi et al., Fast Certified Robust Training with Short Warmup, NeurIPS 2021
> > >
> > > [2] Mueller et al., Certified Training: Small Boxes are All You Need, ICLR 2023
> > >
> > > [3] Mao et al., Connecting Certified and Adversarial Training, NeurIPS 2023
> > >
> > > [4] De Palma et al., Expressive Losses for Verified Robustness via Convex Combinations, ICLR 2024
> > >
> > > [5] Mao et al. Understanding Certified Training with Interval Bound Propagation, ICLR 2024
> > >
> > > [6] Baader et al., Expressivity of ReLU-Networks under Convex Relaxations, ICLR 2024

---

### Official Review · Reviewer_t9Mr · 2024-11-01

**Soundness:** 2
**Presentation:** 3
**Contribution:** 2
**Rating:** 3
**Confidence:** 4

**Summary:**

This paper proposes a library for benchmarking certified training methods under unified settings. It uses the best practices for certified training from (Shi et al., 2021), such as CNN7 architecture with batch normalization, IBP initialization, warm-up schedule and warm-up regularizers. To improve generalization, it uses L1 regularization and stochastic weight averaging (Izmailov et al., 2018). From the implementation perspective, the authors propose to use full batch statistics to address problems with batch normalization when gradient accumulation or PGD attack is performed. The paper claims that the improvements of recent methods in certified training drop significantly compared to older IBP training method under the same settings with proper hyperparameter tuning. Further, the authors analyze different aspects of the training methods: regularization strength, model utilization, loss fragmentation, OOD generalization and shared mistakes.

**Strengths:**

- The paper raises an important question of fairly assessing the algorithmic improvements of recent certified training methods compared to older IBP-based training. Since the evaluation depends on many factors and components, the paper proposes to fix some of them to the best-known ones and to properly tune the rest.
- The writing is clear (except for the presentation of Table 1), the code for benchmarking, and the weights of pre-trained models are provided.
- The analysis of training methods leads to interesting conclusions. Particularly, the relationship between propagation tightness and certified accuracy at larger epsilon, i.e. the absence of correlation, is surprising.

**Weaknesses:**

I believe the **experiments are insufficient** to support the main claims of the paper. Particularly:

1. **Accuracy-robustness tradeoffs are not considered**. Improvements in robustness can be due to decreased natural accuracy, and vice versa [a, b, c]. For example, in Table 1 for CIFAR-10 at 2/255 the implementations of following methods choose a different point at accuracy-robustness tradeoff curve compared to the one in literature, getting higher robustness at the cost of reduced accuracy: CROWN-IBP, SABR, STAPS, MTL-IBP, making claims about technical improvements unsupported. In this regard, the baselines such as ACERT [a], and ACE [b] are missing. Accuracy-robustness tradeoff curves and metrics such as ART-score [a] can be used to capture the improvements in the tradeoff.
2. **Error bars are missing**.  The presented improvements over the results in the literature could be statistically insignificant. For example, the experimental results for CIFAR-10 at 8/255 in paper by Shi et al. (2021) show standard deviation of $\pm0.3$ for certified accuracy and of $\pm0.4-0.7$ for natural accuracy, which makes improvements in both accuracy and robustness in Table 1 for SABR and TAPS within the error of standard deviation.
3. **Training costs are not considered**. Different methods require different amount of computational costs for training, which could be an important factor to consider in benchmarking.
4. **Certification costs are not considered**. Since some certified training methods allow computing tight certified bounds using efficient "online" certification methods, such as IBP (Gowal et al., 2018, Mao et al., 2024), the IBP-based certified accuracy or IBP-based certified radius [a] could also be compared. The cost of test-time verification might be an important factor in choosing a training method.

Since this is a paper proposing a benchmark, it **lacks original** contributions. In terms of evaluation setting, most of the components were already used consistently in previous works.

Smaller comments:
- The main results in Table 1 are hard to parse and analyze due to large amount of numbers to compare. Accuracy-robustness plots could help with improving clarity.
- Due to shared mistakes, the paper claims that "_... there could be an intrinsic difficulty score for each input_". The certified radius of robustness of each point, described in [a, d], could serve as such score. The average certified radius and/or the histogram of radii [d] can be compared in the benchmark. The adaptive training methods can be discussed in this regard.

[a] Nurlanov, Z., Schmidt, F.R., Bernard, F. (2024). Adaptive Certified Training: Towards Better Accuracy-Robustness Tradeoffs. In: Bifet, A., et al. Machine Learning and Knowledge Discovery in Databases. Research Track and Demo Track. ECML PKDD 2024. Lecture Notes in Computer Science(), vol 14948. Springer, Cham. https://doi.org/10.1007/978-3-031-70371-3_8

[b] Müller, M. N., Balunović, M., & Vechev, M. (2021). Certify or predict: Boosting certified robustness with compositional architectures. In International Conference on Learning Representations (ICLR 2021).

[c] Tsipras, D., Santurkar, S., Engstrom, L., Turner, A., & Madry, A (2019). Robustness May Be at Odds with Accuracy. In International Conference on Learning Representations (ICLR 2019).

[d] Bosman, A. W., Hoos, H. H., & van Rijn, J. N. (2023). A preliminary study of critical robustness distributions in neural network verification. In Proceedings of the 6th workshop on formal methods for ML-enabled autonomous systems.

**Questions:**

The main concerns about the experiments are raised in the weaknesses section. If these can be addressed, I would be happy to change my opinion.

---

> ### Author Response · Authors · 2024-11-19
> **Response to $\Rt$**
>
> We are happy to hear that Reviewer $\Rt$ considers our work important, clearly written, leading to interesting and surprising insights into the deterministic certified training. In the following, we address all concrete questions raised by Reviewer $\Rt$.
>
> **Q1: Is accuracy-robustness tradeoff not considered? For example, some algorithms in Table 1 might get better certified robustness at the cost of reduced accuracy. How is this handled by this work (and this field)?**
>
> We would like to note that in practice, this field mostly focuses on improving the *best certified accuracy*, regardless of the drop in *clean accuracy*. More specifically, all recent SOTA works [1,2,3,4] select their best model solely based on the best certified accuracy, which is thus the basis of literature numbers reported in Table 1. This effectively means that accuracy-robustness tradeoff in this field translates to the right-most point one algorithm can get. With this in mind, we remark that based on the current algorithms, one cannot get better certified accuracy than the numbers reported to the best of their implementation/algorithms, regardless of whether they decide to sacrifice more clean accuracy or not. This is also because all these algorithms are trained solely on the objective of certified robustness, but not clean accuracy.
>
> **Q2: Error bars are not provided for Table 1. Could this make the result statistically insignificant?**
>
> Following the discussion above, all numbers in Table 1 are the best numbers one can get to the best of their efforts. In the case of CTBench numbers (our benchmark), we use the same random seed for all algorithms (thus fixing random batches, etc.) and the same training schedule (thus the same training steps), and then perform a thorough hyperparameter tuning for all algorithms separately. This procedure means Table 1 numbers are highly costly, as also pointed out by Reviewer $\Rs$. Simply selecting a different random seed and reusing the same hyperparameter cannot get the same performance, thus reporting error bars means repeating this full procedure multiple times, which is prohibitively expensive. In addition, based on the described procedure, our numbers all represent the best numbers we can get for each algorithm, rather than the result of a random experiment. This highly reduces the variance of the result, as we perform hyperparameter tuning in a search space of size roughly 50, as described in Appendix B.4.
>
> **Q3: Is training and certification cost considered and how?**
>
> We report the complete training and certification cost in Appendix B.6. Regarding training, we fix the number of steps taken by each algorithm (the same training schedule). Regarding certification, we fix the certification algorithm (one of the SOTA complete verifiers, MN-BaB) and the timeout (1000 seconds per sample). Details can be found in Appendix B.3 and B.5.
>
> **Q4: How should Table 1 be parsed? Could accuracy-robustness plots help with clarity?**
>
> Table 1 reports the clean (natural) accuracy, adversarial accuracy and certified accuracy, both in literature and in our benchmarks. We note that, as discussed in **Q1**, all models in literature and in our benchmark are selected solely based on the best certified accuracy. Therefore, adding accuracy-robustness plots is not meaningful for Table 1. In addition, our work is completely in parallel to [5], which develops conclusions about the maximum certifiable radius. We would also like to note that since our certification budget is 1000 seconds per sample at a given perturbation size, searching for the maximum certifiable radius is computationally prohibitive, thus such plots are naturally impossible to create.
>
> **Q5: Could average certified radius be plotted for the studied deterministic certified training methods? Could adaptive training methods improve certified training?**
>
> Following the discussion above, computing maximum certified radius is computationally prohibitive, thus computing average certified radius is also impossible for us. Regarding adaptive training methods, we acknowledge that such approaches may improve certified training, but this is out of the scope of this work, which designs the library and benchmark.
>
> **Reference**
>
> [1] Shi et al., Fast certified robust training with short warmup, 2021.
>
> [2] Müller et al., Certified training: Small boxes are all you need, 2023.
>
> [3] Mao et al., Connecting certified and adversarial training, 2023.
>
> [4] De Palma et al., Expressive losses for verified robustness via convex combinations, 2024.
>
> [5] Nurlanov et al., Adaptive Certified Training: Towards Better Accuracy-Robustness Tradeoffs.

---

> > ### Comment · Reviewer_t9Mr · 2024-11-28
> > **Response to Authors**
> >
> > I appreciate the authors' response. Unfortunately, my concerns remain. In particular, the concerns about statistical significance of the stated improvements, accuracy-robustness tradeoffs and the absence of efficient certification methods in evaluation are not addressed. Also, as mentioned in the original review, the benchmark lack novelty since it uses established techniques from [1].
> >
> > [1]: Shi et al., Fast certified robust training with short warmup, 2021

---

> > > ### Author Response · Authors · 2024-12-01
> > > **Reply to Reviewer $\Rt$ (1/3)**
> > >
> > > We are happy to further address Reviewer $\Rt$’s concerns below:
> > >
> > > **Statistical Significance of Results**
> > >
> > > We appreciate the reviewer’s concern regarding statistical significance. As observed in prior works on certified training, the reported improvements often fall within a similar order of magnitude. For example, [1] reports certified accuracy gains of approximately 1% for MNIST at $\epsilon=0.3$ and 2% for CIFAR-10 at $\epsilon=8/255$, [2] reports improvements of roughly 0.3% for MNIST at both $\epsilon=0.1$ and $\epsilon=0.3$, 1% for CIFAR-10 $\epsilon=2/255$ and 0.15% for CIFAR-10 $\epsilon=8/255$, and [3,4,5] report similar improvements depending on the setting. In our benchmark, improvements for methods such as SABR and MTL-IBP also fall within this range, highlighting consistency with established research. Also note that in previous work it is common to run multiple random seeds and report only the best certified accuracy, underlining that if our improvements were not statistically significant, previous work could have easily achieved similar numbers.
> > >
> > > Moreover, during our hyperparameter finetuning process, we observe stable trends when varying different parameters. For instance, increasing regularization strength or adjusting the robust weight and epsilon-shrinking factor produces predictable changes in certified and clean accuracy. This demonstrates a high signal-to-noise ratio in our experiments, suggesting that our reported improvements are robust.
> > >
> > > However, we acknowledge the importance of explicitly quantifying uncertainty. Unfortunately, due to the high computational cost of these experiments, it is not feasible to fully address this during the rebuttal phase. For instance, training and certifying a single CIFAR-10 network trained with SABR or MTL-IBP requires approximately 2-3 days on a single GPU, while TinyImageNet takes even longer. However, in Tables S1 and S2 below we present all randomness results on the MNIST dataset, given the lower computational costs. We run each Certified Training method independently for 3 times with the same tuned hyperparameters that we originally reported and report the average and standard deviation. We observe that the standard deviation across 3 random seeds is close to or smaller than 0.1 for almost all methods. The results shows that our improvements for most MNIST methods have a statistical significance of more than $3\sigma$.
> > >
> > > **Table S1**: MNIST 0.1 Randomness results
> > > | Method| Source| Nat| Cert |
> > > |--|---|---|---|
> > > |IBP| Literature|98.84 |97.95 |
> > > | | CTBench manuscript|98.87 |98.26 |
> > > | | Average $\pm$ Stdev | 98.86 $\pm$ 0.06 | 98.25 $\pm$ 0.03 |
> > > | CROWN-IBP | Literature|98.83 |97.76 |
> > > | | CTBench manuscript|98.94 |98.21 |
> > > | | Average $\pm$ Stdev | 98.93 $\pm$ 0.01 | 98.17 $\pm$ 0.05 |
> > > |SABR | Literature|99.23 |98.22 |
> > > | | CTBench manuscript|99.08 |98.40 |
> > > | | Average $\pm$ Stdev | 99.15 $\pm$ 0.08 | 98.42 $\pm$ 0.03 |
> > > |TAPS | Literature|99.19 |98.39 |
> > > | | CTBench manuscript|99.16 |98.52 |
> > > | | Average $\pm$ Stdev | 99.20 $\pm$ 0.05| 98.50 $\pm$ 0.04|
> > > | STAPS | Literature|99.15 |98.37 |
> > > | | CTBench manuscript|99.11 |98.47 |
> > > | | Average $\pm$ Stdev | 99.15 $\pm$ 0.04 | 98.38 $\pm$ 0.10|
> > > |MTL-IBP| Literature|99.25 |98.38 |
> > > | | CTBench manuscript|99.18 |98.37 |
> > > | | Average $\pm$ Stdev | 99.16 $\pm$ 0.03 | 98.31 $\pm$ 0.06 |
> > >
> > > **Table S2**: MNIST 0.3 Randomness results
> > > | Method| Source| Nat| Cert |
> > > |--|---|---|---|
> > > |IBP| Literature|97.67 | 93.10 |
> > > | | CTBench manuscript|98.54 |93.80 |
> > > | | Average $\pm$ Stdev | 98.55 $\pm$ 0.02 | 93.82 $\pm$ 0.10|
> > > | CROWN-IBP | Literature|98.18 |92.98 |
> > > | | CTBench manuscript|98.48 |93.90 |
> > > | | Average $\pm$ Stdev | 98.46 $\pm$ 0.03 | 93.84 $\pm$ 0.12 |
> > > |SABR | Literature|98.75 | 93.40 |
> > > | | CTBench manuscript|98.66 |93.68 |
> > > | | Average $\pm$ Stdev | 98.69 $\pm$ 0.03 | 93.64 $\pm$ 0.06 |
> > > |TAPS | Literature|97.94 |93.62 |
> > > | | CTBench manuscript|98.56 |93.95 |
> > > | | Average $\pm$ Stdev | 98.58 $\pm$ 0.03 | 93.90 $\pm$ 0.11|
> > > | STAPS | Literature|98.53 |93.51 |
> > > | | CTBench manuscript|98.74 |93.64 |
> > > | | Average $\pm$ Stdev | 98.69 $\pm$ 0.06 | 93.60 $\pm$ 0.05|
> > > |MTL-IBP| Literature| 98.80 |93.62 |
> > > | | CTBench manuscript|98.74 |93.90 |
> > > | | Average $\pm$ Stdev | 98.75 $\pm$ 0.02 | 93.82 $\pm$ 0.21 |
> > >
> > > We will incorporate this discussion and the tables in the revised manuscript.

---

> > > ### Author Response · Authors · 2024-12-01
> > > **Reply to Reviewer $\Rt$ (2/3)**
> > >
> > > **Accuracy-Robustness Tradeoff**
> > >
> > > We think the reviewer might be unfamiliar with the common practice of certified training and the reasons behind, thus we provide a further explanation and empirical analysis here. The accuracy-robustness tradeoff is a well-studied concept in the field of certified training, but it works very differently to adversarial robustness. We particularly note that a drop in natural accuracy does not necessarily mean an increase in certified accuracy, because increased regularization might reduce the **true robustness** to ease certification with less precise methods (e.g. with IBP), which could be unnecessary for powerful certification methods. This is why the field (and us) all take the highest point in the curve, i.e., the highest certified accuracy, as the sole basis of SOTA. In addition, analyses about the robustness-accuracy tradeoff already exist in the respective papers, thus revisiting the concept in our work would be less meaningful, because we aim to fully release the potential of respective methods rather than study their internal dynamics. For example, [2] provides a thorough analysis in Figure 7, showing how certified and clean accuracy evolve for different values of $\lambda$, and [4] similarly examines this tradeoff for their methods in Figure 1. In both cases we observe that higher regularization (i.e. higher values for $\lambda$ or $\alpha$) indeed improves IBP certifiability of the network, but heavily damages the natural accuracy which in turn also lowers the network’s empirical and certified robustness.
> > >
> > > However, for completeness, we will include certified accuracy versus natural accuracy plots for a subset of methods in the appendix of the final version. While the manuscript itself can no longer be updated during the rebuttal phase, we provide plots at this [anonymous link](https://mega.nz/file/DFJUATKZ#ZXwyFVaNHKGb4QgVqlhtsixnOgLu5Ra9Fm9XV6L7l88). Figures S1 and S2 present the CTBench results from Table 1 in the original manuscript for MNIST 0.1 and CIFAR-10 2/255 respectively. Figure S3 presents a zoom-in of Figure S2 where we show the robustness-accuracy tradeoff for the three best methods under different hyperparameters: SABR, STAPS and MTL-IBP.
> > >
> > > Since our benchmark is only focused on finding the best certified accuracy of each method, the plot in Figure S3 is also focused on the peak region of the robustness-accuracy plots. We particularly analyze this region and observe the same trends as previous work: decreasing regularization (the direction of increasing natural accuracy) from the level of IBP (equivalent to $\lambda=1$ for SABR and $\alpha=1$ for MTL-IBP) also comes with increased robustness, up to an optimal point. Afterwards reducing regularization further increases natural accuracy, but severely hurts certifiability, up to the point where adversarially trained networks (equivalent to $\lambda=0$ for SABR and $\alpha=0$ for MTL-IBP) exhibit much higher natural accuracies, but close to 0 certified robustness even when considering SOTA verifiers.
> > >
> > > **Efficient Deterministic Certification Methods**
> > >
> > > The reviewer’s comments regarding efficient certification methods seem to overlook the current state of the field. Over the past decade, deterministic certification has been a major area of research, culminating in the development of highly precise methods that have been extensively optimized for efficiency [6,7,8].
> > >
> > > Moreover, cheap certification methods such as IBP are very weak when it comes to proving robustness guarantees for networks trained with recent SOTA-certified training methods. For instance, [2] (Figure 7) and [4] (Figure 1) demonstrate that state-of-the-art trained networks exhibit close to zero IBP certifiability, but very high MN-BAB or OVAL-BAB verified robustness. As a result, while IBP-based certification is computationally cheap, its use for evaluating these networks would be uninformative and inconsistent with standard practices.
> > >
> > > It is also important to note that our work does not aim to innovate on certification algorithms. Like all relevant prior work in certified training, we use existing certification methods as-is, without modification. Our focus is on benchmarking certified training techniques, not developing or analyzing certification algorithms.

---

> > > ### Author Response · Authors · 2024-12-01
> > > **Reply to Reviewer $\Rt$ (3/3)**
> > >
> > > **Novelty of the Benchmark**
> > >
> > > The purpose of a benchmark and library is not to introduce novel algorithms but rather to compile and systematize recent and relevant advances in the field. This effort facilitates future development and evaluation of new methods, providing a consistent and reproducible framework.
> > >
> > > While our work (as well as all relevant related work in the past 3 years) builds upon the techniques proposed by Shi et al. (2021) [1], we want to emphasize that the contributions of our benchmark extend far beyond their scope. For instance, our benchmarking setup incorporates a broad range of methods and systematic hyperparameter tuning, which are absent in [1], and our analysis of training dynamics, regularization strength, loss fragmentation, and shared mistakes are novel and provide deeper insights into certified training. Further, as a benchmark paper, we believe that it is essential to keep the core algorithms under examination unchanged. The reviewer seems to believe that the novelty and impact of all benchmarks are limited because they are evaluating existing algorithms, which we find quite confusing.
> > >
> > > Thus, while we acknowledge the importance of [1], our contributions, as a benchmark and library, are distinct and critical for advancing the field.
> > >
> > > **Conclusion**
> > >
> > > We thank the reviewer for their feedback and hope that our clarifications address their concerns. We remain committed to improving the clarity, completeness, and rigor of our work and look forward to incorporating additional analyses in the final version of the paper.
> > >
> > > **References**
> > >
> > > [1] Shi et al., Fast Certified Robust Training with Short Warmup, NeurIPS 2021
> > >
> > > [2] Mueller et al., Certified Training: Small Boxes are All You Need, ICLR 2023
> > >
> > > [3] Mao et al., Connecting Certified and Adversarial Training, NeurIPS 2023
> > >
> > > [4] De Palma et al., Expressive Losses for Verified Robustness via Convex Combinations, ICLR 2024
> > >
> > > [5] Mao et al. Understanding Certified Training with Interval Bound Propagation, ICLR 2024
> > >
> > > [6] OVAL-BAB, https://github.com/oval-group/oval-bab, multiple publications (2017-2021)
> > >
> > > [7] Ferrari et al., Complete Verification via Multi-Neuron Relaxation Guided Branch-and-Bound, ICLR 2022
> > >
> > > [8] Alpha-Beta-CROWN, https://github.com/Verified-Intelligence/alpha-beta-CROWN, multiple publications (2017-2024)

---

> > > > ### Comment · Reviewer_t9Mr · 2024-12-02
> > > > **Response to Authors**
> > > >
> > > > 1. **Statistical Significance of the Results**
> > > >
> > > > - The fact that referenced papers only report a single best number is not an excuse for the proposed benchmark to also lack higher-order statistics, especially given that the improvements are marginal.
> > > > - I think that **tuning hyperparameters for each random seed** is not the correct way to evaluate the training methods.
> > > > - Do you tune hyperparameters on a separate validation set or on the test set? If it is the latter, than there is a high chance the numbers are overfitted and do not reflect real statistically significant algorithmic improvements.
> > > > - Please note that in contrast to the referenced papers, Shi et al (2021) reports mean and standard deviation of the results, and the variation is higher on CIFAR-10 compared to MNIST. As noted in my original review, the stated improvements in both accuracy and robustness in Table 1 for SABR and TAPS are within the error of standard deviation. This point is not addressed properly in the author responses.
> > > >
> > > > 2. **Accuracy-Robustness Tradeoff**
> > > >
> > > > I am familiar with the mentioned works in certified training. I think it is important to consider applicability of the training methods in practice, therefore it is important to measure the accuracy-robustness tradeoff of the evaluated methods. For example, on CIFAR-10 the natural accuracy drops from typical 91% of standard training to 54% to achieve 35% certified accuracy at a fixed epsilon of 8/255. A drop of 37% in classification accuracy makes the approach of pursuing only the best certified accuracy impractical. I appreciate the provided plots in Figure S3 for 3 methods on a single setting of eps=2/255. These plots also show the tradeoff (e.g. for MTL-IBP and STAPS), in contrast to the author claims. I believe on larger epsilon values, the tradeoffs are even more noticeable. I think that the tradeoff plots should try to reach the level of natural accuracy of standard training. For the proposed benchmark paper, all settings and all considered methods should be equally evaluated.
> > > >
> > > > 3. **Efficient Certification Methods**
> > > >
> > > > The efficient certification methods have their own merits and corresponding applications. The comprehensive benchmark should also include the efficiency of the certification into account, given that certified training methods are known to be closely related to the following verification methods. The current setup has the limit of 1000 seconds for each input, which is quite expensive.
> > > >
> > > > 4. **Novelty of Benchmark**
> > > >
> > > > The proposed benchmark uses the setup exactly as in works referred by authors. It is almost the same as the experiments section of a common paper in the domain, proposing a new method. I believe that the hyperparameter tuning does not represent sufficient novelty. My main complain here is that the benchmark does not address any problems with existing evaluations, and simply repeats the same evaluation protocols with different hyperparameters. As recommended in my original review, considering statistical significance of the algorithmic improvements, studying the accuracy-robustness tradeoffs of the training methods, measuring the efficiency of the certification methods that can be applied post-training -- could address some of the problems with the existing evaluations.
> > > >
> > > > ---
> > > > **References:**
> > > >
> > > > - Shi et al. Fast Certified Robust Training with Short Warmup. 2021

---

> ### Author Response · Authors · 2024-12-04
> **Response to $\Rt$**
>
> **Statistical Significance of Results**
>
> We appreciate the reviewer’s concern and agree that additional statistical analysis will strengthen the manuscript. We commit to including this in the revised version.
>
> Regarding hyperparameter tuning, the approach depends on the goal:
> - To study the variance of the best numbers, one would tune for every seed.
> - To analyze sensitivity to hyperparameters, one would use the same hyperparameters across all seeds.
>
> In our work, we use the latter approach, fixing hyperparameters to focus on sensitivity analysis.
>
> As for the results (numbers are provided in the last response), the improvements for SABR and TAPS on MNIST are statistically significant, with SABR achieving improvements of **6 sigma** (98.22 to 98.40 ± 0.03) for MNIST 0.1 and **4 sigma** (93.40 to 93.64 ± 0.06) for MNIST 0.3, and TAPS achieving **3 sigma** in both settings. We note that we are comparing our average performance to the best performance in the literature reported across random seeds, further highlighting the statistical significance.
> For CIFAR-10 at $\epsilon=2/255$, larger standard deviations are offset by more substantial improvements, yielding comparable statistical significance. At $\epsilon=8/255$, improvements across methods remain minimal, consistent with prior work. We will incorporate detailed numbers in the revised manuscript.
>
> We hope this clarification reassures the reviewer about the robustness and significance of our results.
>
> **Accuracy-Robustness Tradeoff**
>
> We thank the reviewer for raising this point but note that the comments appear directed at the broader field of deterministic certified robustness rather than the specific contributions of our work. The tradeoff between natural accuracy and certified robustness is a well-known challenge in this domain, and obtaining robustness for large perturbations in realistic settings is particularly difficult.
>
> Obtaining even empirical robustness inherently requires sacrificing natural accuracy, as shown by methods like EDAC (78.95% natural accuracy for 42.48% empirical robustness on CIFAR 8/255). Achieving near-standard training natural accuracy is possible but leads to almost zero certified accuracy, making evaluation meaningless. Unlike adversarial robustness evaluation, where computational cost is uniform, certification often involves significant computational expense (e.g., 1000 seconds per input) for unverified samples, emphasizing the practical constraints.
>
> In addition, we would like to point out that not all methods exhibit the same tradeoff. For methods such as IBP and CROWN-IBP, there is **no inherent tradeoff** between certified accuracy and natural accuracy, as is the case for methods like SABR and MTL-IBP, which can specifically tune parameters to achieve networks with varying levels of robustness. We note that the discussion is limited to training with robust loss solely, as adopted by **every** certified training method in the literature.
> We also want to mention again that extensive plots analyzing the accuracy-robustness tradeoff have already been provided in previous works, and adding similar plots to our paper would not represent a novelty. Our goal is to benchmark the existing methods systematically, and while we will provide additional plots in the appendix for completeness, we do not believe this addition will significantly change the analysis presented in previous work.

---

> > ### Author Response · Authors · 2024-12-04
> > **Response to $\Rt$ (cont.)**
> >
> > **Efficient Deterministic Certification Methods**
> >
> > We appreciate the reviewer’s input but note that our benchmark evaluates certified training methods, not certification algorithms. While their efficiency is important, it is beyond the scope of this work.
> >
> > Using cheaper methods like IBP would result in **near-zero certified accuracy for techniques like SABR, TAPS, and MTL-IBP** on challenging datasets like CIFAR (small perturbations), as highlighted in their original papers. Employing more computationally expensive methods is necessary to enable meaningful certification and provide a fair assessment of certified training approaches.
> >
> > **Novelty of the Benchmark**
> >
> > We respectfully disagree with the reviewer's assessment, as it seems to focus solely on the benchmarking aspect of our work while overlooking its broader contributions. While benchmarking naturally involves evaluating existing methods, our work goes far beyond this by introducing significant innovations:
> > - **Unified Library**: We provide a comprehensive library consolidating methods for consistent evaluation and ease of use.
> > - **Corrected Implementations**: Errors and inconsistencies in prior work were addressed, leading to more reliable results.
> > - **Well-Tuned Hyperparameters**: Systematic tuning improves performance, setting new baselines.
> > - **Improved Benchmark**: Combining corrections and optimizations, we establish a new benchmark that raises the standard for certified training methods.
> > - **Novel Insights**: Our analysis goes beyond certified accuracy, exploring training dynamics, regularization effects, loss fragmentation, and shared mistakes across methods—offering unique insights critical for advancing the field.
> > By integrating these contributions, we are not only benchmarking but also driving the field forward by providing a robust foundation for future research.

---

### Official Review · Reviewer_Z4Eb · 2024-11-03

**Soundness:** 3
**Presentation:** 4
**Contribution:** 2
**Rating:** 6
**Confidence:** 3

**Summary:**

The paper introduces a benchmark for certified robust training. The goal is to standardize the hyperparameters, training schedules (& other training configurations) between competing methods in certified training. The purported advantages of newer methods are lower when older baselines were given equivalent optimization and testing conditions. The work covers several popular approaches like PGD, IBP, and CROWN-IBP.

**Strengths:**

-- The paper is well-written and typeset well

-- Tackles an important problem in the field: the inconsistent evaluation of different certified training methods. I think the field needed this kind of paper.

-- It's not only a benchmark paper but provides some analysis into certified model behavior in loss fragmentation (showing certified models reduce fragmentation compared to adversarial training), have shared mistake patterns, model utilization metrics, and generalization performance (showing certified training provides benefits for certain types of corruptions).

**Weaknesses:**

-- The novelty of the paper is limited since it's just focused on benchmarking existing methods. Certified robustness is a relatively new field and the field needs methods as much as unifying benchmarks. I do believe the lack of novelty is mitigated to an extent by the analysis provided in Section 5.

-- I wonder about the sustainability of the benchmark since there are other leaderboards for adversarial training (e.g. RobustBench). Others may want to submit their work to an existing leaderboard rather than standardize to adopt your settings.

-- I'm a bit confused about the purpose of the fragmentation experiments. Robust models lead to fewer flipped neurons in the presence of noise, but why should we care? This is after all expected given they are more robust in general to input noise. I believe these experiments may be valuable but the authors should articulate why.

**Questions:**

Some questions I had while reading:

-- Why do methods like TAPS and MTL-IBP achieve better accuracy while deactivating more neurons?

-- Is there a theoretical framework to explain the relationship between neuron deactivation and robustness?

-- Is there a way to understand and leverage the shared mistakes patterns to improve certified training? Or is it natural that mistakes would overlap (similar to how mistakes overlap in natural training)?

---

> ### Author Response · Authors · 2024-11-19
> **Response to $\Rz$**
>
> We are happy to hear that Reviewer $\Rz$ considers our work is well-written, tackles an important problem, needed by the field and provides novel insights into the deterministic certified training. In the following, we address all concrete questions raised by Reviewer $\Rz$.
>
> **Q1: Is the proposed benchmark sustainable? How could others submit their work to the existing leaderboard?**
>
> We would like to note that the primary goal of our work is to develop a benchmark rather than a leaderboard. Benchmarks differ from leaderboards in that a benchmark needs to evaluate in fair and comparable settings, while leaderboards simply take numbers reported in the literature as grounded. Therefore, benchmarks are naturally less sustainable than leaderboards. In addition, as the field advances, the benchmark setting often evolves. For example, we expect the field will make advances towards scalable certified training, and thus in the future, a larger model than the current SOTA architecture may be used in the future benchmarks. Our benchmark represents the current knowledge of the field, and thus we do not expect submissions of new numbers to this benchmark, otherwise it may introduce unfair evaluations. This deviation of benchmarks and leaderboards has been observed in practice. For example, [1] develops a benchmark along with a leaderboard. While their leaderboard is maintained, their benchmark is never updated, as expected. However, this does not imply that a benchmark is unnecessary for the field, as a fair and high-quality evaluation provides more scientific conclusions than a leaderboard which merely draws reported numbers from literature.
> In addition, our provided library makes development of future benchmarks much easier than before, increasing the sustainability of the benchmark.
>
>
> **Q2: Regarding Section 5.1, why do we care if certified models have fewer flipped neurons (less fragmentation)? Is this expected given that the models are more robust?**
>
> Robustness is not directly related to less fragmentation. For example, as shown in Figure 3, the more robust SABR model has more fragmentation than the less robust IBP model. Therefore, robustness does not imply less fragmentation.
>
> Loss fragmentation is of special interest to certified models. This is because the complexity and thus computational costs of certification algorithms highly depend on loss fragmentation. Therefore, understanding loss fragmentation is important for certified training, and our study and insights about how certified training affects loss fragmentation facilitates future work in certified training.
>
> **Q3: Regarding Section 5.3, why do methods like TAPS and MTL-IBP deactivate more neurons (for clean input) but achieve better accuracy (compared to IBP)? Is there a theoretical framework to explain the relationship between neuron deactivation and (certified) robustness?**
>
> While insufficient model utilization (number of neurons activated) strongly affects performance when the model does not have sufficient capacity, high model utilization does not necessarily bring better performance, e.g., in adversarial training, EDAC has better clean accuracy than PGD but less model utilization. This potentially explains why IBP is worse than MTL-IBP in terms of clean accuracy. Regarding the relationship between the number of deactivated neurons and certified robustness, there is some preliminary but not yet complete theoretical framework explaining this. For example, [2] shows that certified training increases propagation tightness, a metric relating to the tightness of certification, and one way to increase propagation tightness is to deactivate more neurons. Intuitively, this is because IBP bounds of deactivated neurons are zero, which matches the exact bounds.
>
> **Q4: Is there a way to leverage shared mistake patterns to improve certified training?**
>
> We believe this is one promising future direction for certified training. As discussed in Section 6, algorithms leveraging this, e.g., curriculum learning, might further improve certified training. These attempts are out of the scope of this work, and we leave them as future work.
>
> **Reference**
>
> [1] Li et al., Sok: Certified robustness for deep neural networks, 2023.
>
> [2] Mao et al., Understanding certified training with interval bound propagation, 2024.

---

> > ### Comment · Reviewer_Z4Eb · 2024-12-02
> >
> > Dear Authors,
> >
> > Thank you for the response. This clarifies my questions. I do think some concerns remain regarding the novelty of the proposed work. However, I was positive on the paper before and will retain my score.
> >
> > Best regards,
> >
> > Reviewer Z4Eb

---

### Official Review · Reviewer_xBxH · 2024-11-10

**Soundness:** 4
**Presentation:** 4
**Contribution:** 2
**Rating:** 5
**Confidence:** 4

**Summary:**

The paper presents CTBENCH, a standardized library and benchmark designed to fairly evaluate certified training algorithms for neural networks, addressing the inconsistency in previous evaluations due to varied training schedules, certification methods, and under-optimized hyperparameters. By testing all algorithms under consistent conditions with tuned hyperparameters, CTBENCH reveals that most certified training methods perform better than previously reported, setting new benchmarks. Through CTBENCH, authors uncover several interesting properties of models training with certified methods.

**Strengths:**

1. The paper proposes a new benchmark for certified robustness methods for image classifiers.
2. Authors implement several prominent certified robustness methods in a unified framework, thereby standardizing the implementations to facilitate future research.
3. Furthermore, authors correct implementation mistakes and perform systematic hyperparameter tuning to fully realize the potential of all methods.
4. Authors present several interesting findings regarding the properties of ceritified robustness methods, for example, models trained using distinct methods have a high overlap in the examples they succeed and fail on, uncovering a sample-specific inherent difficulty level that can be leveraged to improve training. And, these methods can boost OOD generalization for specific corruptions, and hurt generalization for others.

**Weaknesses:**

1. Authors incorrectly state that the benchmark from Li et al. is not up to date as "it reports 89% and 51% best certified accuracy for MNIST epsilon = 0.3 and CIFAR-10 epsilon = 2/255 in its evaluation, respectively, while recent methods have achieved more than 93% and 62%". However, at the time of this review, the numbers on Li et al.'s leaderboard (https://sokcertifiedrobustness.github.io/leaderboard/) are even higher than 93% and 62%, they are 94.02% and 68.2%. Furthermore, the leaderboard toppers are defenses from 2019/2021. It appears that the authors might have pulled their numbers from a stale source.
2. In order to be an improvement over the existing benchmark (of Li et.al.), one important requirement is comprable or improved comprehensiveness. Based on the results in the paper, the proposed benchmark is significantly less comprehensive than Li et. al. on two important directions: (i) number of defenses evaluated, (ii) number of diverse models used during evaluation. While I understand that the proposed work can be made more comprehensive by running more experiments, this is not the case currently and so is worth poining out.
3. Furthermore, as stated in the limitations section, the propsoed benchmark only focuses on deterministic certified robustness in the L_infinity space. Whereas, Li et. al.'s benchmark uses both determinisitc and probabilistic certified methods, and covers all the popularly used norms in literature (i.e., L_1, L_2, L_infinity). Thereby further hurting the comprehensiveness of the proposed benchmark.
4. Some of the findings presented in this paper are expected and already established by prior works (see Questions).
5. The main contribution of the paper is a unified code-based (and benchmark) for promiment certified robustness methods. Even though authors uncover several interesting findings while reproducing and tuning sota methods, the nature of the contributions of this paper are heavily empirical (not enough technical novelty). As such, this paper is much better suited for venues like TMLR that put emphasis on contributions of such nature.

**Questions:**

1. It is already well established by previous works that robustness training increases local smoothness. What is unique about the findings presented in this paper?
2. It is also previously established that adversarially robust trianing methods tend to have higher sample complexity, and therefore are more likely to overfit (less regularization). Other than the choice of metric, what is unique about the findings in Section 5.4.?
3. Is there an explanation for why the model performs worse for certain corruptions? How will these results be affected if we use different L_p norms? For example, I would expect a model trained to be robust in the L_2 space to be better resistant to Gaussian noise and less resistant to salt and pepper noise.

---

> ### Author Response · Authors · 2024-11-19
> **Response to $\Rx$ (part 1)**
>
> We are happy to hear that Reviewer $\Rx$ feels that our unified library, implementation correction, systematic study and our novel findings into the deterministic certified training are useful and facilitate future research. In the following, we address all concrete questions raised by Reviewer $\Rx$.
>
> **Q1: Authors incorrectly state that the benchmark from [1] is not up-to-date; at the time of this review, the number on their leaderboard website is updated. Does this mean that the authors might have pulled their numbers from a stale source?**
>
> We would like to distinguish a benchmark with a leaderboard. Benchmarks differ from leaderboards in that a benchmark needs to evaluate in fair and comparable settings, while leaderboards simply take numbers reported in the literature as grounded. In fact, other than a leaderboard which simply draws reported numbers from the publications, [1] also provides a benchmark study, as can be seen from their website. This benchmark is not updated and is what we referenced, reporting 89% best certified accuracy for MNIST $\epsilon=0.3$.
>
> On the other hand, their leaderboard is updated by the original authors, collecting reported numbers from the literature. This means that this leaderboard is mixed: different algorithms use different architecture, particularly different activations. This makes this leaderboard unfair in the sense that different algorithms are not directly comparable. To solve this, we consider all algorithms based on ReLU networks rather than specialized activations, as this matches the wide practice of deep learning. We pull our literature numbers from a series of the most recent SOTA publications [4,5,6], representing the best practices in the field.
>
>
> **Q2: Improved comprehensiveness over [1] is expected. However, this work only evaluates deterministic certified training while Li et al. evaluates certification methods and deterministic/randomized certified training methods on different norms. Does this mean the contribution of this work is covered/shaded by [1]?**
>
> We would like to note that [1] is a SoK paper providing a meta-analysis about the general certified robustness area, while our work develops a library and benchmark about deterministic certified robustness. Therefore, our contributions are in nature not directly comparable to theirs in comprehensiveness, as they focus on comprehensiveness while we focus on an in-depth study regarding deterministic certified robustness. Readers should not expect the same coverage in our study, as we are not meta-analyzing the field. In contrast, we provide an easy-to-use library for deterministic certified training, while [1] did not provide such toolboxes. In addition, the benchmark in [1] naturally does not cover the most recent advances, and thus cannot draw insights about the current progress, while our study achieves both goals. Therefore, our contribution is by no means covered by [1].
>
>
> **Q3: Previous works have shown robust (adversarial) training increases local smoothness; what is unique about the findings presented in Section 5.1?**
>
> Section 5.1 observes certified training induces less loss fragmentation. While this relates to increased local smoothness, they are not equivalent. For example, models with the same level of loss fragmentation could have different local smoothness. In addition, instead of studying smoothness, we focus on loss fragmentation for a specific reason in certified training: the complexity of certification algorithms highly depends on loss fragmentation, but not on local smoothness. This is why understanding loss fragmentation is important for certified training, but local smoothness is not directly related, to the best of our knowledge. Therefore, Section 5.1 is of specific interest to the deterministic certified training community, which is the main goal of our study.

---

> ### Author Response · Authors · 2024-11-19
> **Response to $\Rx$ (part 2)**
>
> **Q4: Previous works have shown adversarial training tends to have a higher sample complexity and overfit more easily (less regularization); what is unique about the findings presented in Section 5.4?**
>
> Adversarial training is known to overfit easily, called robust overfit. We would like to note this is not directly related to less regularization, as adversarial training naturally puts more regularization than standard training, e.g., increasing local smoothness. The nuance here is that not all regularization relates to overfitting; instead, each regularization simply introduces an inductive bias into the model. For example, while $L_2$ regularization which asks for small weights usually prevents overfitting, we could also ask for large weights as a special inductive bias. Therefore, investigating different regularization has its own value.
>
> Our study focuses on one special inductive bias (regularization) which is closely related to certified training: propagation tightness [7]. Basically, high propagation tightness makes certification easier for loose relaxations, but not necessarily for tight relaxations. Therefore, understanding propagation tightness introduced by different certified training algorithms helps the field to gain insights and develop future algorithms.
>
> **Q5: Regarding Section 5.5, is there an explanation about why certified models perform worse for some corruptions? Could different training norms affect this?**
>
> This is a very good question. Since all corruptions we studied are out-of-distribution and not directly controlled by certified training, we do not know the exact answer about why some corruptions appear to be more difficult. Different training norms might affect this, and we agree that intuitively training with regard to the $L_2$ norm might resist Gaussian noise better. However, we would like to note that currently the field of deterministic certified training focuses on $L_\infty$ norm and other norms are largely overlooked. In particular, no well-performing $L_2$-norm deterministic certified training algorithm has been developed. Therefore, it is out of the scope of our work to investigate this question.
>
> **Reference**
>
> [1] Li et al., Sok: Certified robustness for deep neural networks, 2023.
>
> [2] https://sokcertifiedrobustness.github.io/leaderboard/
>
> [3] Lyu et al., Towards Evaluating and Training Verifiably Robust Neural Networks, 2021.
>
> [4] De Palma et al., Expressive losses for verified robustness via convex combinations, 2024.
>
> [5] Müller et al., Certified training: Small boxes are all you need, 2023.
>
> [6] Mao et al., Connecting certified and adversarial training, 2023.
>
> [7] Mao et al., Understanding certified training with interval bound propagation, 2024.

---

### Author Response · Authors · 2024-11-19
**General Response**

$\newcommand{\Rx}{\textcolor{green}{xBxH}}$
$\newcommand{\Rz}{\textcolor{blue}{Z4Eb}}$
$\newcommand{\Rt}{\textcolor{purple}{t9Mr}}$
$\newcommand{\Rs}{\textcolor{orange}{s9m2}}$

We thank all reviewers for their insightful reviews, helpful feedback, and interesting questions. We are particularly encouraged to hear that reviewers consider our library and benchmark to be important ($\Rx$, $\Rz$, $\Rt$, $\Rs$), our insights novel and useful ($\Rx$, $\Rz$, $\Rt$, $\Rs$) and our paper well-written ($\Rz$, $\Rt$, $\Rs$). No shared concern from reviewers is identified, thus we will address all concrete questions in individual responses.

---

### Meta-Review · Area_Chair_QNPa · 2024-12-20

**Metareview:**

This paper develops a benchmark and library for several representative deterministic certified training algorithms for fair comparison on hyperparameters, training schedules and (exact) certification methods. The reviewers generally agree that this paper is well-written, however, lacking technical novelty due to the nature of the topic. The submission will benefit from incorporating error bars of the results on all the dataset reported to show informative comparison. On a side note, the L2 (and more generally, Lp, p>=1) certified training can be directly extended from the Linf certified training method, since propagating the bounds are essentially the same and the only difference would be at the input when using holder's inequality. The authors are urged to include other Lp results to further strengthen the paper.

**Additional Comments On Reviewer Discussion:**

During the rebuttal period, reviewers and authors have fruitful discussions regarding the novelty, technical contributions, statistical significance of the results, and the impact of the work. I agree with the reviewers that there are several unresolved problems (e.g. lacking statistical significance on the numbers reported in the table, the authors should report results using multiple exact/inexact verifiers, and investigate Lp certified training results).

---

### Decision · Program_Chairs · 2025-01-22

Reject